# Climate-induced storminess forces major increases in future storm surge hazard in the South China Sea region

Melissa Wood[1], Ivan D. Haigh[1], Quan Quan Le[2], Hung Nghia Nguyen[2], Hoang Ba Tran[2], Stephen E. Darby[3], Robert Marsh[2], Nikolaos Skliris[2], Joël J.-M. Hirschi[4], Robert J. Nicholls[5], and Nadia Bloemendaal[6]

[1] School of Ocean and Earth Science, National Oceanography Centre Southampton, University of Southampton, Waterfront Campus, European Way, Southampton, UK; *Correspondence to: Ivan Haigh (I.D.Haigh @soton.ac.uk)*

[2] Southern Institute of Water Resource Research (SIWRR), 658th Vo Van Kiet Avenue, Ward 1, District 5, Ho Chi Minh city, Vietnam

[3] School of Geography and Environmental Science, University of Southampton, Highfield, Southampton, UK

[4] National Oceanography Centre Southampton, University of Southampton, UK

[5] Tyndall Centre for Climate Change Research, University of East Anglia, Norwich, UK

[6] Institute for Environmental Studies (IVM), Vrije Universiteit Amsterdam, 1081 HV, Amsterdam, The Netherlands

**Submitted to Natural Hazards and Earth System Sciences; December 2021**

## Abstract

Coastal floods, driven by extreme sea level, are one of the most dangerous natural hazards. The people at highest risk are those living in low-lying coastal areas, exposed to tropical cyclone forced storm surges. Here we apply a novel modelling framework to estimate past/present and future storm surge and extreme sea level probabilities along the coastlines of south China, Vietnam, Cambodia, Thailand, and Malaysia. A regional hydrodynamic model is configured to simulate 10,000 years of synthetic tropical cyclone activity, representative of a past/present (1980-2017) and a high-emission scenario future (2015-2050) period. Results show that extreme storm surges, and therefore total water levels, increase substantially in the coming decades, driven by an increase in the frequency of intense tropical cyclones. Storm surges along the south China and northern/southern Vietnam coastlines increase by up to 1 m, significantly larger than expected changes in mean sea level rise over the same period. The length of coastline that is presently exposed to storm surge levels of 2.5 m or greater more than doubles by 2050. Sections of Cambodia, Thailand and Malaysia coastlines are projected to experience storm surges (at higher return periods) in the future, not previously seen, due to a southward shift in tropical cyclone tracks. Given these findings, coastal flood management and adaptation in these areas should be reviewed for their resilience against future extreme sea levels.

## 1. Introduction

Around the world's coastlines it is estimated that ~230 million people are directly exposed to some level of storm surge hazard from either tropical or extra-tropical cyclone activity (SwissRe, 2017). The populations most acutely

at risk from storm surge induced extreme sea levels are those settled on low-lying coastlines within tropical zones associated with intense tropical cyclone (TC) activity (McGranahan et al., 2007; Woodruff et al., 2013; Kirezci et al., 2020; Edmonds et al., 2020; Dullaart et al., 2021). Yet global assessments of flood risk regularly overlook the contribution of low probability TC events when considering storm surge induced extreme sea level flooding (Muis et al., 2016; Dullaart et al. 2021). Furthermore, while there is considerable uncertainty regarding future changes in TC intensity and frequency, particularly at a local scale, it is thought that the risk of TC induced storm surge flooding will increase in the future (Bloemendaal et al., 2022). To better protect present and future coastal communities, it is vital that we improve our understanding of local-scale TC driven storm surge hazard and risk.

The main difficulty in assessing TC induced storm surge hazard, both at present and into the future, is that intense TC events are, by their very nature, infrequent (Mori et al., 2019; Dullaart et al., 2021). TCs are not only rare events, but they typically affect comparatively short stretches of coastline (<500km) as they approach land, and so storm surges are often under-represented in the data from the sparsely distributed network of global tide gauges (Pugh and Woodworth 2014; Bloemendaal et al., 2020). Furthermore, analysing extreme storm surge behaviour, and estimating storm surge hazard, ideally requires long (50-100 years) time series of sea level data, which do not exist in most tropical regions (Irish et al., 2011). This limitation is acutely problematic because extreme sea level probabilities based on short records are notoriously imprecise with large uncertainties (Irish et al., 2011; Lin and Emanuel 2016; Kirezci et al., 2020).

To overcome these problems, previous studies have adopted two different approaches with the same goal of extending the historic or predicted future sea level record available from existing tide gauge data. The first approach is to reconstruct multi-decadal storm surge and/or total water level time-series through the use of statistical models to infer surge time-series from more widely available meteorological datasets. These methods use simple linear or multiple regression models on climate indices to reconstruct long time series of surge levels from which extreme values and trends can be more robustly estimated. This has been done at both regional and global scales, using, for example, the tide gauge record and the 20[th] Century Reanalysis dataset (Cid et al., 2018, 2017; Zhang and Wang 2021) or a mixture of climate reanalyses data (Wahl and Chambers 2016; Tadesse and Wahl 2021). Statistical approaches mostly benefit from modest computational resource needs, but this advantage is traded-off against the use of meteorological forcings that often have insufficient spatial resolution in tropical regions to capture the effects of cyclone activity on sea levels (Haigh et al., 2014; Cid et al., 2018).

The second approach involves the use of hydrodynamic models to generate multi-decadal time-series of storm surge and extreme sea levels across oceanic domains. TC induced storm surges are challenging to model at continental or global scales because these intense storms typically have diameters less than the model mesh resolution or are smoothed out in the large grid cells of meteorological datasets, meaning they are therefore difficult to resolve (Murakami and Sugi, 2010; Larson et al., 2014; Takagi et al., 2017; Bloemendaal et al., 2019b; Kirezci et al., 2020). An earlier version of the Global Tide and Surge Model (GTSM; using ERA-Interim reanalysis data – see Dee et al., 2011) was found to underestimate TC induced extreme sea levels for this reason. This problem was subsequently improved in the latest GTSM iteration, v3 (Muis et al. 2020), with an updated model resolution and use of the ERA5 reanalysis meteorological dataset (Hersbach, 2020). The authors successfully simulated past/present storm surges and extreme sea levels.

To address the scarcity of adequate low probability storm surge events within extreme sea level analysis, several studies have recently attempted to force numerical storm surge models with synthetic datasets that seek to represent long-term TC activity over many hundreds to thousands of years. For example, Haigh et al. (2014) extended the work of Harper et al., (2009) and generated a 10,000-year synthetic dataset of TC activity for the Australian region, which thus included extreme TCs larger than in the observational dataset but that were physically plausible. This 10,000-year atmospheric dataset was used to force a MIKE 21 hydrodynamic model of the Australian coast to estimate present-day sea level exceedance probabilities more accurately. More recently Bloemendaal et al. (2020) similarly developed a synthetic TC dataset called STORM (Synthetic trOpical cyclone geneRation Model) which statistically resampled and simulated TC tracks and intensities from 38 years of historical atmospheric data from the International Best Track Archive for Climate Stewardship dataset (IBTrACS; Knapp et al., 2010) to the equivalent of 10,000 years under the same climate conditions. Dullaart et al. (2021) subsequently coupled this STORM data with the GTSMv3 model to produce past/present (1980-2018) storm surge and extreme sea level return period estimates for coastlines world-wide, directly tackling the problems of precision in relative location (of tide gauges) and availability of TC storm surge data. The earlier studies, for example of Haigh et al. (2014) and Dullaart et al. (2021), focused on past/present day extreme sea levels and didn't apply the approach to look at possible future changes in TC activity with climate change.

Looking to the future, coastal flood hazard is expected to increase; primarily due to rising relative mean sea level, but also due to changes in storm surges driven by changes in the frequency, intensity, and tracks of tropical and extra-tropical cyclones (Kirezci et al., 2020) and change in tides (Haigh et al., 2020). Note, wave setup and run up is also likely to be important in some regions, but as discussed later, for simplicity we ignore the influence of waves in this paper. To date, a large number of studies have focused on assessing changes in global mean sea levels. Much less research has been devoted to determining the contribution of climate-driven changes in storm activity in forcing extreme storm surges (Fox-Kemper et al., 2021). While there is consensus that there will be substantial changes to the frequency and severity of tropical (and extra-tropical/mid-latitude) cyclones in the future (Mousavi et al., 2011; Woodruff et al., 2013; Wahl et al., 2017; Knutson et al., 2020; Emanuel 2021) the two most recent reports of the Intergovernmental Panel on Climate Change (IPCC) underscore that there is currently '*low confidence*' (~ 20% chance) in our ability to correctly predict how climate-driven storm surges may contribute to changes in future sea level extremes (Wong et al., 2014; Fox-Kemper et al., 2021). This deep uncertainty arises not only from the small number of studies assessing changes in storm surge available at the time of the last IPCC Assessment Review, but also because of the significant challenge of predicting changes in tropical (and mid-latitude) cyclone activity at a local and regional scale.

Studies to date have also assumed that storm surge extreme behaviour has been, and will continue to be, stationary in the future (Hinkel et al., 2014; Vitousek et al, 2017). But with projections of a changed climate by the end of this century, this hypothesis has been challenged in recent modelling studies (Lin-Ye et al., 2020; Tadesse and Wahl, 2021). For European coastlines by 2100, modelling shows that extreme storm surge levels may augment relative sea-level rise by over 30%, under mean SSP5-8.5 climate projections (Vousdoukas et al., 2016). More recently Calafat et al. (2022) examined tide gauge observations from 1960 to 2018 for north-western European seas and discovered statistically significant past changes in storm surges extremes due to climate variability and anthropogenic forcing, over this period. This trend has already affected the likelihood of surge extremes in this region and is likely to be magnified in the future. Since existing coastal flood defences were originally designed

under the presumption of stationary surge extremes, there is a compelling question about how effective current coastal flood defences actually are now against present storm surge hazard, in this region and beyond. Not only today, but the issue has strong implications for future coastal planning too. Thus, it is vital we accurately assess past and future changes in storms surges and extreme sea levels, especially in areas projected to experience a shift in intense TC activity under our changing climate.

Therefore, the overall aim is this paper is to more accurately estimate both present and future storm surge and extreme sea level probabilities along coastlines in areas with intense TC activity. As a case study, we focus here on the densely populated coastline of south China, Vietnam, Cambodia, Thailand, and Malaysia in south-east Asia (Fig. 1). South-east Asia has long been identified as a 'hotspot' for projected future mean sea level rise, plus extremes of sea level related to TC activity (McGranahan et al., 2007; Nicholls and Cazenave, 2010; Kirezci et al., 2020; Nicholls et al., 2021). Of the most densely populated metropolitan areas at risk from storm surges around the world, 8 out of 10 are located in south-east Asia (SwissRe, 2017). And 4 of those 8 are located within our proposed study area (Ho Chi Minh City [14], Pearl River delta [near 4], Shantou [near 3] and Manila [near 26] in Fig.1). The study area sits within the Western North Pacific (WNP) TC region, which currently accounts for almost one-third of all TC counts globally (Gray 1975, 1977). WNP TCs (typhoons in this region) are projected to become more intense over the course of the 21st century, with higher category cyclones increasing in frequency (Chan, 2005; Zang and Church 2012; Emanuel 2013; Woodruff et al., 2013; Lap, 2019; Knutson et al., 2020; Emanuel 2021; Bloemendaal et al. 2022). Within the study area, a key interest is the coastline of Vietnam, as this is the area of focus of the project that funded this study. More than 70% of Vietnam's population lives in coastal regions (GFDRR, 2015), with a large proportion residing in one of its two deltas - the Red and Mekong River deltas (Fig.1). These delta communities are especially vulnerable to flooding because the low-lying land is densely populated, rich in infrastructure and high value assets, with a river able to funnel storm surges further inland.

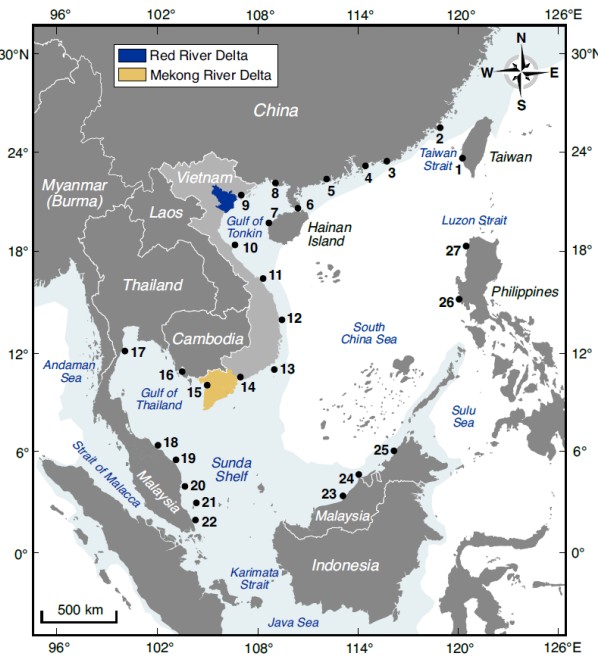

*Figure 1 – South China Sea model domain, with location of tidal gauges numbered (also see Table 1) and the location of the Red and Mekong River Deltas highlighted. The shaded blue area, in-sea, shows the approximate coverage of the continental shelf, at ~250 m depth.*

The specific objectives and structure of the paper is as follows. As a first objective, we configure a depth averaged hydrodynamic model of the South China Sea and extensively validate it against measured sea level data from tide gauges in the region. A description of the hydrodynamic model and validation exercise is provided in Sect. 2. As a second objective, we force the hydrodynamic model with 10,000 years of TC activity for the past/present (1980-2017) and future (2015-2050; based on a high-emission scenario), from the novel STORM synthetic dataset of

Bloemendaal et al. (2020, 2022). From the model outputs, we estimate both past/present and future storm surge and extreme sea level probabilities along the coastlines of south China, Vietnam, Cambodia, Thailand, and Malaysia. The approach we take to simulate the storm surges and calculate the associated extreme probabilities is described in Sect. 3. As a third objective, we compare the past and future probabilities, first for just the storm surge component and then for total water levels (i.e., storm surge plus astronomical tide). The results of this

comparison are described in Sect. 4. As a sub-objective, we also examine the tracks of the cyclones that are responsible for generating the largest storm surges in particulate locations along the coastline of the case study area. The key findings are discussed in Sect. 5, and conclusions are given in Sect. 6.

## 2.   The hydrodynamic model configuration and validation

We start this section off by describing the configuration of the hydrodynamic model (Sect. 2.1). We then discuss

the model validation against observations, first for the astronomical tidal component (Sect. 2.2) and then for storm surges and total water levels (Sect. 2.3).

### 2.1 Model configuration

We configured a depth-averaged (i.e., barotropic) hydrodynamic model of the South China Sea using the Danish Hydraulic Institute's, MIKE 21 FM (flexible mesh) suite of modelling tools (DHI, 2017a). The MIKE 21 FM

model uses the solution of the incompressible Reynolds averaged Navier–Stokes equations, utilising the assumptions of Boussinesq and of hydrostatic pressure. The spatial discretization of the primitive equations is performed using a cell-centred finite volume method. The MIKE 21 FM model uses an irregular triangular mesh to represent the domain area. This provides computation efficiencies by optimising element size to range between coarse resolution for deep ocean, to a more precise representation around detailed coastlines or for features of

interest. We describe each step of the model configuration below. In addition, a schematic giving an overview of the model configuration steps and data inputs is provided in Appendix 1, Fig. A1.

The model grid we created for the South China Sea is shown in Fig.2a. It extends from approximately 100 degrees W to 120 degrees W and 3 degrees S to 25 degrees N. It encompasses the east coast of Malaysia and Thailand, the entire coast of Cambodia and Vietnam, and extends to South China. We developed the model off the

continental shelf of these countries with the eastern domain of the model running along the coastline of Borneo Malaysia, Brunei, the Philippines, and Taiwan (Fig.1). The model grid has seven open sea tidal boundaries, shown in red on Fig.2a. The model mesh reduces from ~52 km at the open sea boundaries to approximately ~2 km along parts of the coast. Along our study coastline (the grid cells shown in green in Fig.2a), the model mesh reduces in resolution from ~11 km along south China and Malaysia to ~5 km around Hainan Island and along the coast of

Thailand and China, down to ~2 km along the Vietnam coastline (as mentioned above, we focus in on Vietnam, as this is the central region of interest in the project that funded this study). We are aware that some global hydrodynamic models (such as GTSMv3; Muis et al. 2020) now have a resolution along the coast of finer than 2

km, and we could have increased the coastal resolution further. However, we purposely didn't go to a finer resolution because: (1) our study involved running the model almost one hundred thousand times for synthetic cyclones - increasing the resolution would significantly increase total run-time; and (2) our model design should consider our Vietnamese co-authors' objective to potentially use the model for future studies, without having regular access to a super-computer. Therefore, we wanted the model to be easy to run on a standard desktop computer.

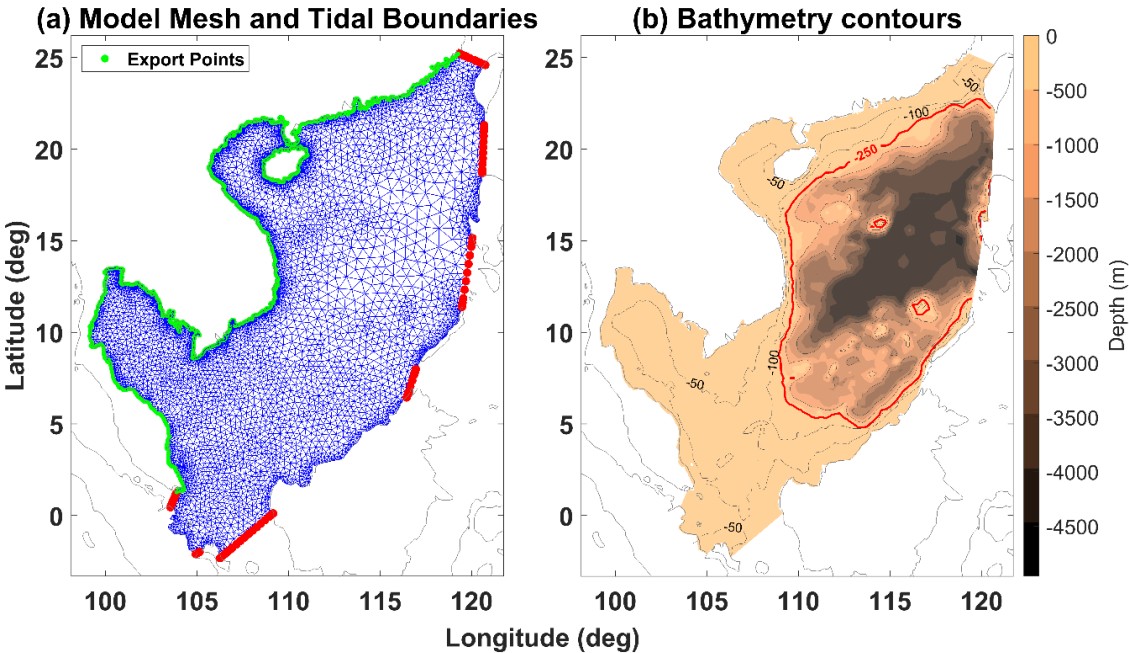

Figure 2 – (a) MIKE 21 FM model mesh for the South China Sea. The irregular triangular mesh grid (blue) of the model has tidal boundaries shown in red. The coastline points exported from the model for analysis are also shown (green). (b): SRTM15+ ocean bathymetry for our South China Sea domain, showing nearshore -50m and -100m contours as well as the -250m depth contour (red) approximating the edge of the continental shelf.

To define the coastal land boundary of the model mesh we used the Global Shoreline dataset from the National Geospatial Intelligence Agency, obtained via the National Oceanic and Atmospheric Administration (https://shoreline.noaa.gov). For bathymetry, we used the 15 arcseconds resolution global dataset from SRTM15+ (v2). This bathymetry data was downloaded from the Scripps Institution of Oceanography website (https://topex.ucsd.edu; Tozer, et al., 2019) and interpolated onto the model grid. All model data therefore is measured using an EGM96 vertical reference datum. The interpolated model bathymetry is shown in Fig.2b.

To generate the astronomical tidal component, we forced the open model boundaries with tidal levels derived from the Oregon State University Tidal Inversion Software (OTIS), TPXO (Egbert & Erofeeva 2002; Martin et al., 2009). The tidal harmonic constituents were downloaded from the OTIS web site (http://volkov.oce.orst.edu/tides/) for points along the seven open sea boundaries of our model, using the OTIS regional China Seas and Indochina region model at 1/30 degree resolution. The data are provided for eight primary (M2, S2, N2, K2, K1, O1, P1, Q1), two long period (Mf, Mm) and three non-linear (M4, MS4, MN4) harmonic constituents. With these data, the tide was then predicted, for each boundary grid point, using the Tidal Model Driver (TMD) MATLAB toolbox (http://polaris.esr. org/ptm_index.html), which has been designed specifically

to predict tidal levels from OTIS formatted harmonic constituents. We also accounted for direct gravitational forcing in the model simulations by including the 'tidal potential' forcing incorporated within MIKE 21 FM (DHI, 2017a). Note, we forced the model with astronomical tides for the validation exercise (see Sect. 2.2 and 2.3), but when running the TC simulations (Sect. 3) we ran surge only simulations, switching off the tidal forcing.

For bed friction the MIKE 21 FM model uses the Manning's formula, and we used the model's default roughness coefficient value 32 $m^{1/3}s^{-1}$. The overall discrete model timestep was set to 3,600 seconds (i.e., 1 hourly), however the MIKE 21 FM hydrodynamic module which resolves the shallow water equations over the model domain uses a variable timestep to ensure stability of the model during the simulation. This means that timesteps would reduce further (between 0.01 and 25 seconds) as necessary, between outputs, to ensure optimal time integration and space discretization solutions. For all other settings (e.g., eddy viscosity, etc) we use the default MIKE 21 FM settings. A schematic of the basic model configuration is also provided in Appendix 1, Fig. A1.

## 2.2 **Model validation of astronomical tides**

We undertook two validation exercises to ensure that our MIKE 21 FM hydrodynamic model accurately captures the complex tidal, storm surge and total water level (tide plus storm surge) characteristics of the study region. In this section we describe the first validation, which focuses on just the astronomical tidal component (Appendix 1, Fig. A2[2d]). The South China Sea has complex tidal characteristics, with some regions experiencing semi-diurnal tides, some mixed, and other regions experiencing strong diurnal, with varying tidal ranges (Phan et al., 2019).

We obtained measured sea level data, at hourly frequency, at 27 tide gauge stations located around the South China Sea model domain, from the University of Hawaii Sea Level Centre (https://uhslc.soest.hawaii.edu; Caldwell, et al., 2015). The locations of the tide gauge sites are shown numbered in Fig.1 and listed in Table 1. Extra years of sea level data at four of these tidal gauge sites (Phu Quoc, Phu Quy, Son Tra, and Rach Gia) was also made available directly from the Southern Institute of Water Resource Research in Vietnam. Despite there being a good number of tide gauges around the South China Sea region, only a third have 30 or more years of data, and several sites have very short records (Table 1). To remove the major meteorological influences and extract just the astronomical component, and to overcome the problem of an incomplete data record at some tide gauge locations, we undertook a harmonic analysis on the available observed levels using the MATLAB T-Tide tidal analysis software (Pawlowicz et al. 2002). We obtained the standard set of 67 tidal constituents, for the most recent year with the least amount of missing data. We then used MATLAB T-Tide software again, with the extracted harmonic constituents, to predict the tide, hourly, for January 2019. We chose the year 2019, because 2019 had the most completed measured records across the 27 sites. At each tide gauge site, we calculated the annual mean sea level value in 2019, and then subtract this level from the data, to offset each time-series so it was equivalent to the model datum of MSL.

*Table 1 - Validation of tidal levels output by the model. Mean absolute difference errors (Mean Absolute Error - MAE) between modelled and observed tide gauge data, for January 2019, at each gauged location in Fig.1. The standard deviation around this MAE is also given.*

| Tide Gauge | ID (Fig. 1) | Latitude (degrees) | Longitude (degrees) | Date range and [number of years of data available] | Mean absolute error (MAE, m) | Standard deviation of MAE (m) | Correlation Coefficient |
|---|---|---|---|---|---|---|---|
| Kaohsiung | 1 | 22.61 | 120.28 | 1980-2016 [37] | 0.07 | 0.05 | 0.95 |
| Xiamen | 2 | 24.42 | 118.30 | 1954-1997 [28] | 0.29 | 0.20 | 0.97 |
| Shanwei | 3 | 22.65 | 115.30 | 1975-1997 [23] | 0.10 | 0.08 | 0.95 |
| Hong Kong | 4 | 22.27 | 114.38 | 1962-2018 [33] | 0.13 | 0.11 | 0.93 |
| Zhapo | 5 | 21.50 | 111.78 | 1975-1997 [23] | 0.12 | 0.09 | 0.97 |
| Haikou | 6 | 20.02 | 110.28 | 1976-1997 [22] | 0.24 | 0.17 | 0.77 |
| Dongfang | 7 | 19.10 | 108.62 | 1975-1997 [23] | 0.16 | 0.12 | 0.94 |
| Beihai | 8 | 21.48 | 108.98 | 1975-1997 [23] | 0.20 | 0.13 | 0.98 |
| Hon Dau | 9 | 20.67 | 106.82 | 1995 [1] | 0.32 | 0.25 | 0.89 |
| Vung Ang | 10 | 18.18 | 106.35 | 1996-1997 [2] | 0.21 | 0.14 | 0.81 |
| Son Tra | 11 | 16.10 | 108.22 | 2009 [1] | 0.09 | 0.07 | 0.96 |
| Qui Nhon | 12 | 13.77 | 109.38 | 1994-2018 [22] | 0.16 | 0.13 | 0.85 |
| Phu Quy | 13 | 10.52 | 108.93 | 2008-2009 [2] | 0.20 | 0.14 | 0.87 |
| Vung Tau | 14 | 10.34 | 107.01 | 1980-2018 [39] | 0.18 | 0.12 | 0.97 |
| Rach Gia | 15 | 9.99 | 105.07 | 1996-2018 [23] | 0.12 | 0.09 | 0.84 |
| Phu Quoc | 16 | 10.22 | 103.97 | 2008-2009 [2] | 0.08 | 0.06 | 0.91 |
| Ko Lak | 17 | 11.79 | 99.90 | 1985-2018 [34] | 0.15 | 0.09 | 0.93 |
| Geting | 18 | 6.25 | 102.12 | 1986-2015 [30] | 0.13 | 0.08 | 0.86 |
| Cendering | 19 | 5.26 | 103.23 | 1984-2015 [32] | 0.16 | 0.11 | 0.92 |
| Kuantan | 20 | 3.97 | 103.44 | 1983-2015 [33] | 0.19 | 0.12 | 0.93 |
| Tioman | 21 | 2.81 | 103.60 | 1985-2015 [31] | 0.18 | 0.13 | 0.93 |
| Sedili | 22 | 1.93 | 104.18 | 1986-2015 [30] | 0.19 | 0.13 | 0.90 |
| Bintulu | 23 | 3.45 | 113.03 | 1992-2015 [24] | 0.13 | 0.09 | 0.95 |
| Miri | 24 | 4.39 | 113.90 | 1992-2014 [23] | 0.08 | 0.06 | 0.97 |
| Kota Kinabalu | 25 | 5.98 | 116.07 | 1987-2015 [29] | 0.14 | 0.11 | 0.93 |
| Subic Bay | 26 | 9.75 | 118.30 | 2007-2018 [12] | 0.07 | 0.06 | 0.96 |
| Currimao | 27 | 14.76 | 120.00 | 2009-2018 [10] | 0.05 | 0.04 | 0.97 |

We then ran the hydrodynamic model for January 2019 (including 2 days of warming period prior to the start of January 2019), forcing the model at the boundary conditions with the OTIS-derived tidal level and no meteorological forcing. Hourly results were output for the model grid points located closest to the 27 tide gauge sites. The resulting time-series, of model-simulated (red line) and observed tide levels (blue line), are shown in

Fig.3 for six select gauge sites along the Vietnamese coastline (sites 10 to 14, Fig.1). It is clear at these six locations, and the other sites (not shown), that the model does an excellent job reproducing the complex tidal dynamics of the study area. For example, the model effectively captures the transition from diurnal tides along the northern Vietnam Coast (sites 10-13) to semi-diurnal tides, with larger range around the Mekong Delta in southern Vietnam (site 14), and then back to smaller diurnal tides on the west side of the south coast of Vietnam

(site 15). Overall, the observed tidal characteristics are captured well across all 27 sites.

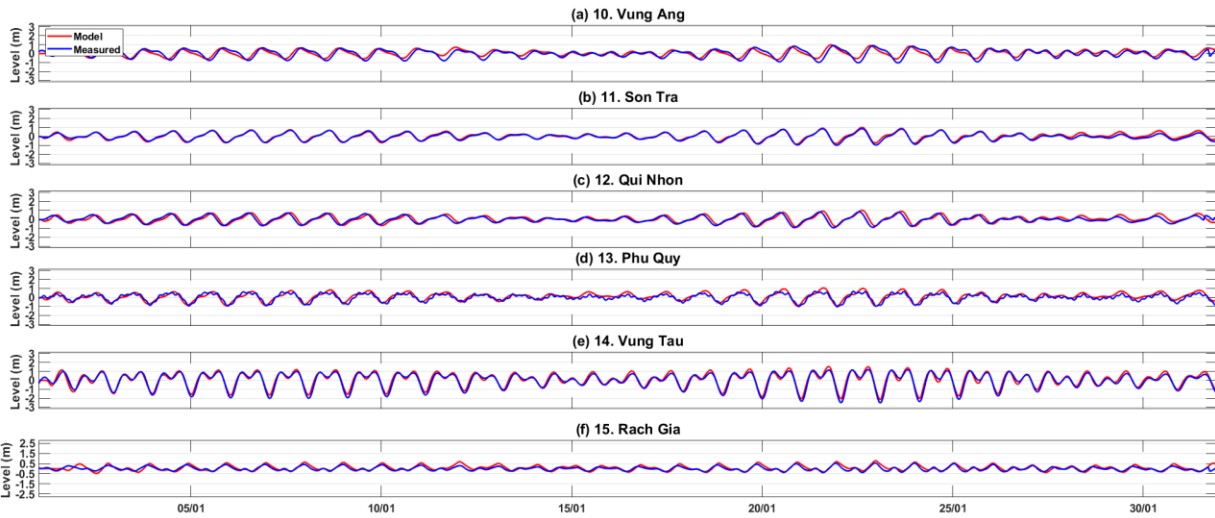

*Figure 3 - Comparison of modelled (red) and measured (blue) astronomical tidal time series (January 2019) at six Vietnamese tide gauge station locations (see Fig.1 for locations).*

To quantify the difference between measured and predicted tides, we calculated the Mean Absolute Error (MAE), the standard deviation around this MAE, and the correlation coefficient between time-series, for all 27 tide gauge sites. These validation statistics are listed in Table 1. Across all 27 sites, the average MAE is 0.15 m, and the mean standard deviation of the MEA is 0.1 m. The parts of the model with the largest MAE errors are typically located at sites with diurnal tides, around the Gulf of Tonkin (sites 6-9 in Fig.1), and the smallest errors are for sites with semi-diurnal to mixed tides around Vietnam, Borneo, and the southern China coastlines (sites 1-5, 10-16, 18-26 in Fig.1). The correlation coefficients range between 0.77 and 0.98 with the highest correlations in the northern and eastern areas of the model that experience fully- or mainly- semi-diurnal tidal regimes. The magnitude of differences between the measured and predicted tide is consistent with other past hydrodynamic modelling studies (e.g., Haigh et al., 2014; Muis et al., 2016; Vousdoukas et al., 2016) and highlights again that the model does accurately reproduce tidal characteristics across the model domain.

At each of the 27 tide gauge sites we also computed and compared the amplitude and phase differences of the four main tidal constituents, extracted from the model-simulation and measured time-series using T-TIDE. The mean absolute amplitude and phase errors of the four main tidal constituents, averaged across all 27 tide gauge sites, are listed in Table 2. The mean absolute amplitude error of the four main tidal constituents, are 0.05, 0,03, 0.06 and 0.06 m, for $M_2$, $S_2$, $O_1$ and $K_1$, respectively. There is a slight amplitude underestimation where there are transitioning tidal regimes, such as the amplitude of larger semi-diurnal tides around the Taiwan strait (Xiamen) and mixed diurnal tides around the Gulf of Tonkin. The mean absolute phase error of the $M_2$, $S_2$, $O_1$ and $K_2$ constituents are 17, 18, 11 and 12 degrees, respectively. Small semi-diurnal ($M_2$ and $S_2$) phase differences exist in the model for tide gauges located around the mixed (mainly diurnal) tide zones of central Vietnam. Phase and amplitude errors may be due to the absolute decimal accuracy of some tide gauge location coordinates as much as due to model limitations. Overall, the results from this validation exercise shows that the model is accurately reproducing tidal characteristic (both in terms of form and range, and individual constituents) across the study domain.

*Table 2 - Mean absolute amplitude and phase errors of the four main tidal constituents for the 27 validation tide gauge sites.*

| Tidal Constituent | Mean absolute amplitude error (m) [s.d.] | Median absolute phase error (degrees) [s.d.] |
|---|---|---|
| $M_2$ | 0.05 [0.04] | 16.8 [15] |
| $S_2$ | 0.03 [0.02] | 18.4 [17] |
| $O_1$ | 0.06 [0.06] | 11.3 [7] |
| $K_1$ | 0.06 [0.06] | 11.8 [9] |

## 2.3 **Model validation of storm surges and total water levels**

In this section we describe the second validation exercise, in which we assess the model's ability to accurately simulate storm surges induced by TCs, and corresponding total water levels (astronomical tide plus storm surge). The length of measured sea level data available, as already indicated and illustrated in Table 1, is on average short

across the 27 gauge sites located within this study area. Consequently, only a small selection of large storm surge events are represented in the available tide gauge records. We therefore focused on those select past cyclone events for validation. A schematic of the process is given in Appendix 1, Fig. A2[1].

The first step was to identify potential significant TC-driven storm surge events in the South China Sea that we could simulate, and for which we could derive wind and atmospheric fields to force the model. To do this we used

data from the IBTrACS version 4 TC database (https://www.ncdc.noaa.gov/ibtracs/; Knapp et al., 2010). We collated all the TCs in IBTrACS, for the WNP region and for the period 1970 to 2020, which: (1) made landfall; (2) have matching measured sea level data at a tide gauge close to the landfall location; and (3) capture the storm surge in the measured records for that event. Unfortunately, radius to maximum winds information was only available in the IBTrACS data for certain cyclones and this therefore further reduced the possible number of TCs

we could use for validation (as we require information on radius to maximum winds to drive an empirical wind and atmospheric pressure model, see below). Only four TC events with significant measured storm surges matched the above criteria, namely: (1) Typhoon Sally in September 1996; (2) Tropical Storm Linda in October/November 1997; (3) Typhoon Ketsana in September 2009; and (4) Typhoon Mangkhut in September 2018. These cyclones impacted different stretches of coastline and thus provided a range of events suitable for validation of the model.

To simulate these four TC events, the second step was to create spatially and temporally varying wind and atmospheric pressure fields to force the hydrodynamic model. We forced the model with two different meteorological fields and compared the results. First, we used $u$ and $v$ wind and MSL atmospheric pressure fields directly from the ERA5 reanalysis data set (Hersbach et al., 2020). These were downloaded from the Copernicus climate data store (https://cds.climate.copernicus.eu/), for the known cyclone dates, on a regular 0.25° x 0.25°

grid at hourly resolution (Hersbach et al., 2020). These data were simply clipped to the area of interest and imported into a MIKE 21 FM grid file format with no other modification. As discussed in the introduction, ERA5 may not accurately capture the intensity and track of TCs, due to its spatial resolution. Hence, we also derived a second set of meteorological fields for each of the four chosen storms. In this instance, we derived spatially and temporally varying wind and atmospheric pressure fields using the empirical approach of Holland (1980). To do

this, we used the Cyclone Wind Generation toolbox (DHI, 2017b) built within MIKE 21 FM. To generate the empirical wind and pressure fields we inputted the track, of each of the four selected TCs at 3-hourly timesteps, as captured in the IBTrACS database, along with central atmospheric pressure and radius to maximum wind values. We selected the 'Single Vortex Holland' option within the toolbox, which creates an estimate of the

Holland B parameters using the Holland Formula specified in Harper and Holland (1999). This tool therefore
generated $u$ and $v$ wind, and pressure files, for each of the four TC events, on a 0.25° x 0.25° grid resolution to match ERA5 spatial grid resolution, for fair comparison.

The third step was to run the model to simulate behaviour for each of the four TC events. For each TC, we ran the model three times. First, we ran the model forced with the wind fields derived from ERA5 and then second, forced with the empirical wind fields derived from the Holland model. For each TC, we also ran the model a third time, with just astronomical tidal forcing at the open boundaries (i.e., no meteorological forcing) so that we could isolate the storm surge components in the wind-forced simulations. A schematic of this storm surge validation procedure is given in Appendix 1, Fig.S2. For each simulation, we ran the model for approximately 5 days around the event; this included two days of warm up, a day before the event, the day of the event and a day after the event. We then finally, for each of the four TCs, visually compared the simulation results against the measured record, at the tide gauge closets to where the TC made landfall. To quantify the difference between predicted and measured total water levels and storms surges, we calculated the MAE between modelled and measured time-series for these four TC events (Table A2, Appendix 1).

A plot of model-simulated (red line) and observed (blue line) total water level, and storm surge only, time-series are shown in Fig.4a and 4b, respectively, for TC Ketsana. TC Ketsana made landfall close to the tide gauge at Son Tra, Vietnam (site 11 in Fig.1) in September 2009, producing a storm surge of approximately 1.4 m. Results for the other three TC events are shown in Appendix 1 (Figures S4, S5 and S6). For each of the four TC events used in this validation, it is clear than the simulations driven with ERA5 wind data significantly under-estimate the magnitude of the storm surge, and hence total water levels. However, for all four TC events, the model simulations driven with wind and pressure forcing derived from the empirical Holland model, do a better job to reproduce the overall magnitude of the storm surge and total water level. The MAE for each of the four TC events' total water level and storm surge are similarly listed in Table A2 in Appendix 1 (for both the simulations driven with ERA5 meteorological fields, and the simulations driven by the empirical Holland model results). When the ERA5 derived forcing fields are used the MAEs vary between 0.17 m and 0.26 m for storms surge, and 0.24 m and 0.32 m for total water level, but reduce to 0.08 m to 0.17 m for storm surge and 0.15 m to 0.29 m for total water level when the empirical forcing fields are used in the model. Overall, these validation findings provide confidence that the hydrodynamic model is able to accurately capture both total water levels and the storm surge component of TC events, when the empirical Holland meteorological forcing approach is used to generate wind and pressure fields for the hydrodynamic model.

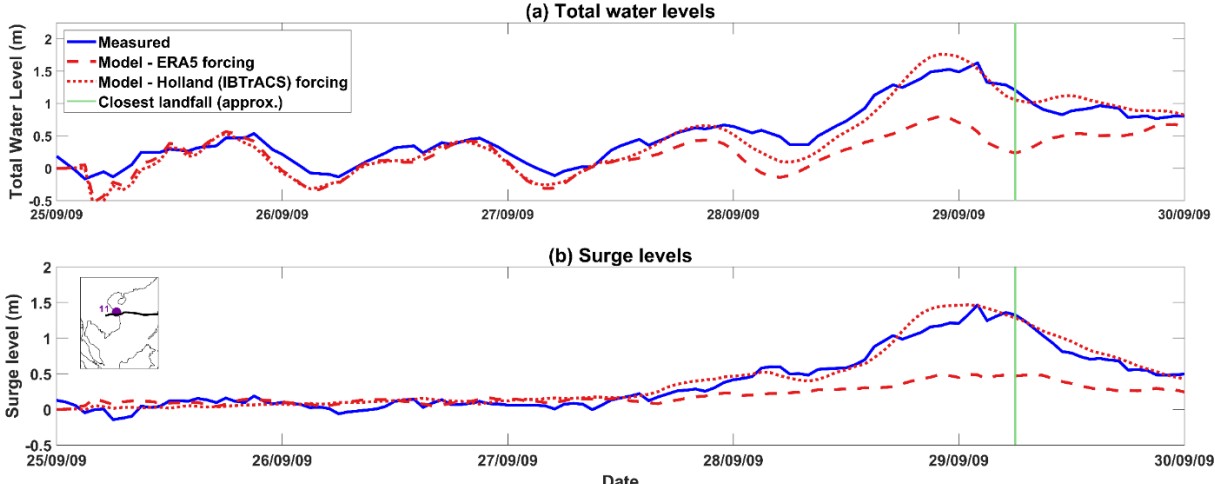

*Figure 4 – (a) measured and modelled total water levels, and (b) measured and modelled storm surges at Son Tra (site 11 in Fig.1) for Typhoon Ketsana, which made landfall at approximately 6am UTC on 29th September 2009 (green vertical line). Modelled total water level and surges using ERA5 (red dashed) and Holland Model (red dotted) wind and pressure fields against measured data (blue).*

## 3.  Approach for simulating present and future extreme sea levels

In this section we start by describing the 10,000-year TC database, representative of the past/present and future high emission scenario (Sect. 3.1). Then we detail how we force the validated hydrodynamic model with these 10,000-year datasets of TC activity (Sect. 3.2). Finally, we discuss how we estimate both past/present and future storm surge and extreme sea level probabilities along the coastlines of south China, Vietnam, Cambodia, Thailand, and Malaysia from the model simulations (Sect. 3.3).

### 3.1 **Synthetic tropical cyclone datasets**

To estimate extreme storm surges and total water levels along our study coastline, we utilised synthetic TC data from the STORM database. Bloemendaal et al. (2020) developed and applied the STORM algorithm to TCs from 38 years of historical IBTrACS data (1980–2018) to statistically extend the original record into the equivalent of 10,000 years of TC activity and create the original past/present STORM database. The STORM dataset preserves

the TC statistics found within the original 38-year dataset. STORM was developed to mimic the seasonality of the observed data it uses, so for TCs in south-eastern Asia genesis occurs between May and November. The STORM database therefore provides 3-hourly, seasonally appropriate information on an individual cyclone's location, wind speed, pressure, radius to maximum winds and storm category. Further details, including a link to download the data itself, is available in Bloemendaal et al. (2020).

We extracted all TCs in the WNP area that reached at least hurricane strength (Category 1 or greater on the Saffir-Simpson Hurricane Wind Scale; Simpson and Saffir, 1974) from the global STORM dataset (Bloemendaal et al. 2020). In the WNP, this amounts to 156,879 individual synthetic cyclones. We then excluded TCs that didn't cross our model domain or those that were very short lived (lasting < 9 hours). This left a sub-set of 30,843 individual TC for our study area which we henceforth refer to as the baseline data sub-set.

For the future datasets we utilized the future STORM synthetic TC dataset of Bloemendaal et al. (2022). They extended their original study by applying the STORM algorithm to extracted data from four high-resolution climate models, namely, CMCC-CM2-VHR4, CNRM-CM6-1, EC-Earth3P-HR, and HadGEM3-GC31-HM.

Each climate model was originally run at a high spatial resolution for the period 2015-2050 and forced with emissions representative of the SSP5-8.5 high emission climate change scenario. The SSP5-8.5 climate change scenario represents unconstrained growth in economic output and energy, which exploits abundant fossil fuel resources and relies on global markets and technological progress to achieve sustainable development (Pielke 2022; IPCC, 2019). It describes a society that develops within the highest greenhouse gas emissions pathway, linked to greater reliance on adaptation, rather than mitigation, to address climate challenges. Using a so-called 'delta approach' Bloemendaal et al. (2022) utilized TCs extracted from the high-resolution climate model runs to statistically generate synthetic events representative of 10,000 years of TC activity, for each of the four future climate simulations. Results from the study indicate that the probability of intense TCs, on average, more than doubles in most regions, including the WNP. Further details, including a link to download the future STORM datasets from a repository, can be found in the Bloemendaal et al. (2022). To see if this pattern is replicated within our study area, Table 3 gives the number of TCs in each of the four future STORM datasets, that pass within the bounds of our model domain, as well as the number of TCs that pass within the bounds of the model domain for the (unfiltered) baseline STORM dataset. The table shows the number of tropical storms or depressions and category 1, 2, 3, 4 and 5 TCs in each option. The numbers do indeed reflect the global trend; of a decrease overall in the number of tropical storms and depressions, but an increase in the number of more intense (category 1-5 on the Saffir-Simpson scale) TCs in the future. Although, as expected, there are differences in numbers of TCs between the STORM datasets derived from the four different future climate models also.

Due to the large computation expense of simulated the equivalent of 10,000 years of TC activity (and within the constraints of the budget of the study that funded this work) it was not possible to submit MIKE 21 FM model simulations for all four future climate datasets. Hence, we elected to only use data from a single future global climate model run. We selected the future STORM dataset based on the CNRM-CM6-1 climate model run because it approximately matched the average number of most intense (category 4 and category 5) TCs across all four climate model options. We thus extracted all future synthetic TCs from the CNRM-CM6-1 climate run, using the same filtering procedure as for the baseline dataset above (to additionally remove the short-lived TCs). This left a sub-set of 63,328 individual TCs for our study area which we henceforth refer to as the future data sub-set.

Heatmaps illustrating the resulting track density of the synthetic (filtered, sub-set) TCs passing through the bounds of the model domain, for the baseline period (representative of the period 1989-2018) and future period (representative of the period 2015-2050 with CNRM-CM6-1, SSP5-8.5 climate scenario), are shown in Figures 5a and 5b, respectively. The difference between the two track densities is shown in Fig.5c. One interesting observation is that the density of TC tracks shift southward in the future scenario (Fig.5b), resulting in a greater number of category 1-5 TCs impacting the central Vietnam coastline in the 35-year interval period up to 2050, and for the first time a density of TCs will reach more southern coastlines too (Fig.5c). Equivalent heat maps were created for the remaining climate models shown in Table 3, as shown in Appendix 1 (Section 4, Fig. A8) for comparison. These also show a projected increase in category 1-5 TCs impacting the central Vietnam coastline over the coming decades.

*Table 3 – Number of baseline and future TCs of each category (using Saffir Simpson scale) within the reduced area of the model domain.*

| Dataset | Tropical Storm/ Depression | 1 | 2 | 3 | 4 | 5 |
|---|---|---|---|---|---|---|
| STORM Past/Present | 54,255 | 29,114 | 14,410 | 13,003 | 5,938 | 120 |
| (a) STORM Future: CNRM-CM6-1 | 31,450 | 33,595 | 20,196 | 19,113 | 14,762 | 924 |
| (b) STORM Future: EC-Earth3P-HR | 34,018 | 35,000 | 19,736 | 18,254 | 12,375 | 657 |
| (c) STORM Future: HadGEM3-GC31-HM | 30,409 | 33,322 | 20,923 | 20,202 | 13,213 | 491 |
| (d) STORM Future: CMCC-CM2-VHR4 | 37,685 | 36,672 | 18,553 | 15,911 | 10,598 | 621 |

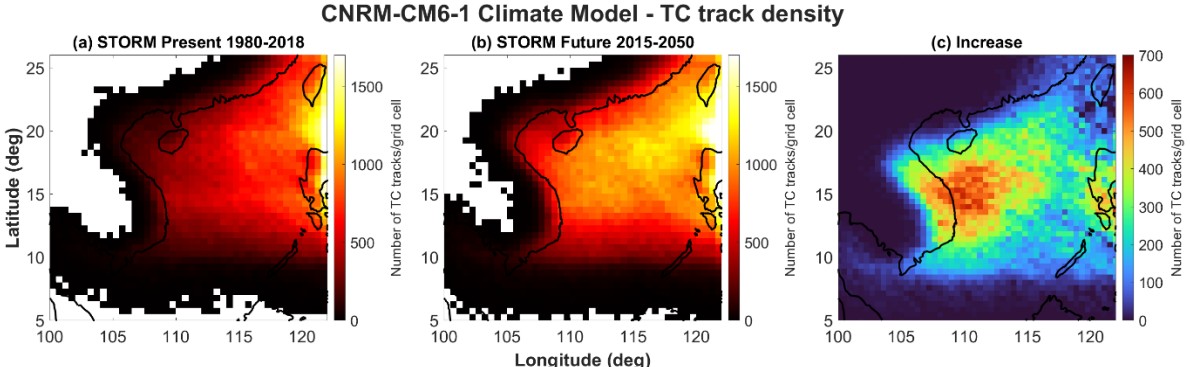

*Figure 5 - Track density = the number of Saffir Simpson category 1-5 TC tracks passing through each 0.5 ° x 0.5 ° grid cell within each ~35 year period. Left: Baseline STORM Past/Present track density of Saffir Simpson Category 1+ (i.e., excluding Tropical Storms), Middle: CNRM-CM6-1 climate model- STORM Future track density, of Saffir Simpson Category 1+, and Right: The cyclone track density difference between them.*

## 3.2 **Hydrodynamic model implementation**

We now describe how we forced the model with wind and pressure fields derived from these 30,843 baseline and 63,328 future TCs. We generated spatially and temporally $u$ and $v$ wind and atmospheric pressure fields, using the approach described above in Sect. 2.3. We used the MIKE 21 FM Cyclone Wind Generation toolbox, inputted with the track of each synthetic TC at 3-hourly timesteps, along with central atmospheric pressures and radius to maximum winds values, obtained from the STORM dataset. Again, we selected the 'Single Vortex Holland' option, with Holland B parameters estimated using the Holland Formula specified in Harper and Holland (1999), and generated $u$ and $v$ wind and pressure files on a 0.25° x 0.25° resolution grid. This spatial resolution was sufficient to resolve the TC within these forcing files, especially as the wind/pressure files would be further interpolated in the MIKE 21 FM software to the higher resolution of the model mesh as the cyclone traverses through the model domain.

For each individual baseline and future TC, we then simulated storm surge levels using the validated MIKE 21 FM hydrodynamic depth-averaged model, described above. We ran each simulation separately, from when each synthetic TC started, or entered the model grid domain, to when it dissipated, or exited the model domain. We decided not to run each simulation with astronomical forcing, instead we just simulated the storm surge component. To justify this choice (which was also followed in many previous studies, e.g., Dullaart et al., 2021) we had previously run a series of sensitivity tests to check if there are significant non-linear interactions between

the astronomical tide and storm surge components which would influence model output of still total water levels in our study region (Horsburgh and Wilson, 2007; Idier et al., 2019). Flood hazard can be underestimated if these non-linear interactions are not accounted for (Williams et al., 2016; Arns et al. 2020). Our sensitivity tests are described in detail in Sect. 3 of Appendix 1 and show that non-linear interactions between tide and surge are indeed negligible in this region. Consequently, we ran 'surge-only' simulations. We ran each model simulation on the University of Southampton's IRIDIS 5 High Performance Computing Facility. On average, each separate TC simulation took around 15 minutes to complete. For reasons of data economy, we chose to only save predicted storm surge time-series, at 10-minute temporal resolution, for each TC, at discrete points along the study area coastline only. As a result, we save model outputs for 3,051 coastline model grid points located along the length of the Chinese, Vietnamese, Cambodian, Thai, and Malaysian coastlines in our study domain (green points in Fig.2a). We automated this whole process. The final output was time-series of storm surges at 3,051 coastal grid points, for each of the 30,843 baseline, and 63,328 future, TCs.

### 3.3 Computation of return periods

Upon completion of all the simulations, the final stage in the analysis was to estimate storm surge and extreme sea level probabilities (i.e., return periods), along the study area coastline, representative of the past/present baseline (1980-2018; 30,843 TCs) and future (2015-2050; 63,328 TCs) period. To estimate return periods of extreme storm surges, we employed the following methodology, based on the approach of Haigh et al. (2014). First, for each of the 3,051 coastal grid points we calculated the annual storm surge maxima, for both the baseline and future data sub-sets. We were able to do this, because the STORM database assigns a synthetic year (from 1 to 10,000) for each TC. On average, there are between 2 and 15 TCs making or approaching landfall within our domain area each STORM year. Note, we used the annual maximum method, as opposed to a peaks over threshold approach, because our interest is in selecting surge peaks from TC events which are independent of one another (a potentially unsafe assumption for these STORM datasets). Choosing annual maxima data more likely secures this independence requirement, and 10,000 years of annual maxima still provides plentiful data to work with. Second, the annual maxima storm surge levels were then sorted in descending order and given a rank ($m$). Third, then the probability of exceedance ($P$) was calculated using the following Gringorten formula:

$$ P = \frac{(m-a)}{(n+2a)} \tag{1} $$

where $a$ is scale parameter equal to 0.44, and $n$ is the number of annual maxima observations. The storm surge return period is given as 1/P. The Gringorten formula was used due to its suitability for extreme value estimation (irrespective of sample size) and previous record in unbiased return period estimation (Guo, 1990). Hence, for each coastal grid point, we calculated storm surge return periods for both the past baseline and future datasets.

We then calculated return periods for total water sea level return for each of the 3,051 coastal grid points. Because non-linear interactions between tide and storm surge were determined to be negligible for this region (see Sect. 3 in Appendix 1), we did this by adding surge levels to a randomly selected astronomical tide, accounting for seasonality in TC, and repeating the process in a Monte Carlo framework. The first step was to run a tide-only model simulation, for the year 2019, and save the predicted tidal levels at each of the 3,051 coastal grid points, at 10-minute intervals. For each coastal grid point, in a second step, we then run a harmonic analysis on the modelled

time-series using the MATLAB T-Tide packaged (Pawlowicz et al., 2002). Using the computed harmonic constituents, we predicted the tides over a longer 19-year period (2003 to 2021), for each coastal grid point. A full 19-year period was used because it encompasses a complete 8.85-year cycle of lunar perigee and 18.6-year lunar nodal cycle, both of which can influence extreme water levels (Haigh et al., 2011; Peng et al., 2019; Baranes et al., 2020).

The third step was then to select a random date from this 19-years of tide data, accounting for seasonality in TCs. Each synthetic TC in STORM has an assigned month. In our study region, the baseline and future synthetic TCs developed largely between May and November, as occurs in the observed record for this region. Because of this it was possible to match each TC (surge) to the correct month in the tide data, preserving TC seasonality. We subsequently allocated each TC a random time, day, and year from the 19-year tidal cycle. For each individual TC, we extracted the height of the astronomical tide, at this given time, day, month, and year, and combined it with the predicted corresponding storm surge. We then repeated the steps outlined above - i.e., computed annual maximum total water levels at each grid point, ranking these and computing extreme total water level probabilities using the Gringorten formula. To account for uncertainty, we repeated the process 100 times in a Monte Carlo approach, combining each storm surge value with a different tidal level. Hence, for each coastal grid point, this produced 100 total water RPL estimates. From these we calculated an average RPL, for each grid point, and for the baseline and future datasets. The 95[th] percentile confidence intervals around the mean value were also calculated, for baseline and future scenarios, and are shown in our results below.

## 4. Results

In this section we compare the past and future return periods, first for just the storm surge component (Sect. 4.1) and then for total water levels (i.e., storm surge plus astronomical tide; Sect. 4.2). We also briefly examine the tracks of the TCs that are responsible for generating the largest storm surges in particular locations along the coastline of the case study area (Sect. 4.3).

### 4.1 Extreme storm surge return period levels

First, we focus on the storm surge only results. Return period levels were estimated as described in Sect. 3.3 from the storm surge heights output by the model, in metres above the MSL. The computed 10% annual exceedance probability (AEP) (1 in 10 year), 1% AEP (1 in 100 year) and 0.1% AEP (1 in 1000 year) return period levels (RPLs) for the baseline past/present scenario (1980-2018) are shown in Figures 6a, 6b and 6c, respectively. The figures show the model grid point output RPLs along the study coastlines of south China, Vietnam, Cambodia, Thailand, and Malaysia. As expected, the storm surge RPLs are lowest along the coastlines of Malaysia and Thailand, which don't typically experience TCs. They then increase moving northwards along the coastline of Cambodia and Vietnam. Storm surge RPLs are highest along the south China coast, reaching values of up to 3.5 m, corresponding with greater frequency of TCs in this area. The shape of the coastline is also shown to have a strong modulating effect on surge RPL, that is especially noticeable for the more extreme events, whereby the modelled surges are typically amplified within the many bays, river mouths and inlets located along this northern coastline (Jelesnianski, 1972). Another effect of the shape of the shore is seen along Vietnam's central coastline, where storm surge RPLs are substantially lower (e.g., 1% AEP surge levels there average ~0.7 m) compared to

the coastlines of north and south Vietnam (1% AEP surge levels average ~1.6 m and ~1.1 m, respectively). The narrow width of the continental shelf in central Vietnam (Fig.1) acts to reduce surge amplitude; behaviour that is noticeable even for the most extreme surges. The correlation between storm surge height and continental shelf width is a well-documented characteristic (e.g., Pugh and Woodworth, 2014).

The computed 10%, 1% and 0.1% AEPs, for the future scenario (2015-2050), representing the SSP5-8.5 high
emission climate change scenario, and from the CNRM-CM6-1 climate model run - are shown in Figures 6d, 6e and 6f, respectively. The spatial patterns observed along the coast closely resemble those of the baseline dataset (Figures 6a, 6b and 6c), but the RPLs are elevated by up to ~1 m in places. The differences between past and future scenario are shown in Figures 6g, 6h and 6i for the 10%, 1% and 0.1% AEPs, respectively. The increase in surge RPLs is largest along the south China coast, and along the northern and exposed southern Vietnamese coast.
For the 1% AEP level, the increase over time is approximately 0.8 m around the northern and exposed southern Vietnam coastline (Fig. 6h). The shape of the coastline, specifically a wide and gently sloping continental shelf and the angle of cyclone approach contribute to this amplification of surge RPLs around these particular coastlines, notably including around the more vulnerable Red and Mekong River deltas in Vietnam (Fig.1; Poulose et al., 2018; Bloemendaal et al., 2019b; Pandey and Rao 2019; Ramos-Valle et al., 2020). The greatest 0.1% AEP
surge level increase exceeds 1.5 m along the Chinese coastline (Fig.6i). For the 0.1% AEP, there are also large differences for the coastlines of Cambodia, Thailand, and the northern part of Malaysia. The baseline model outputs indicate that the coastlines of Cambodia, Thailand and Malaysia are currently relatively unaffected by storm surges linked to the lower category TCs. This is expected, as these coastlines have historically rarely experienced TC induced storm surges of magnitude >10% AEP. For more probable storm surge events up to the
10% AEP standard, this relatively unaffected status is predicted to continue into the future too (Fig.6d). Present-day 10% AEP storm surge heights along these coastlines average around 0.36 m - and between today and mid-century there appears to be zero increase in levels. However, going to more extreme storm surge probabilities in the future scenarios, sections of this coastline are projected to experience storm surges up to 0.6 m (1% AEP) and 0.8 m (0.1% AEP) higher than current levels (Fig.6h and 6i). In some locations this doubles the current (baseline)
storm surge heights.

Looking at the entire study coastline again, the length of coastline that is exposed to 1% AEP storm surge levels of 2.5 m (~95[th] percentile), or greater, more than doubles - going from 353 km to 930 km total length, between the baseline and future scenarios. For the more extreme 0.1% AEP outcome, the baseline scenario has approximately 231 km of coastline with a surge RPL of 3.5 m (~95[th] percentile), or greater. Mostly seen in south
China. This length increases in extent, in the future scenario, to around 577 km of coastline - extending into neighbouring coastline in south China and for the first time including sections of north and south Vietnam coastlines too.

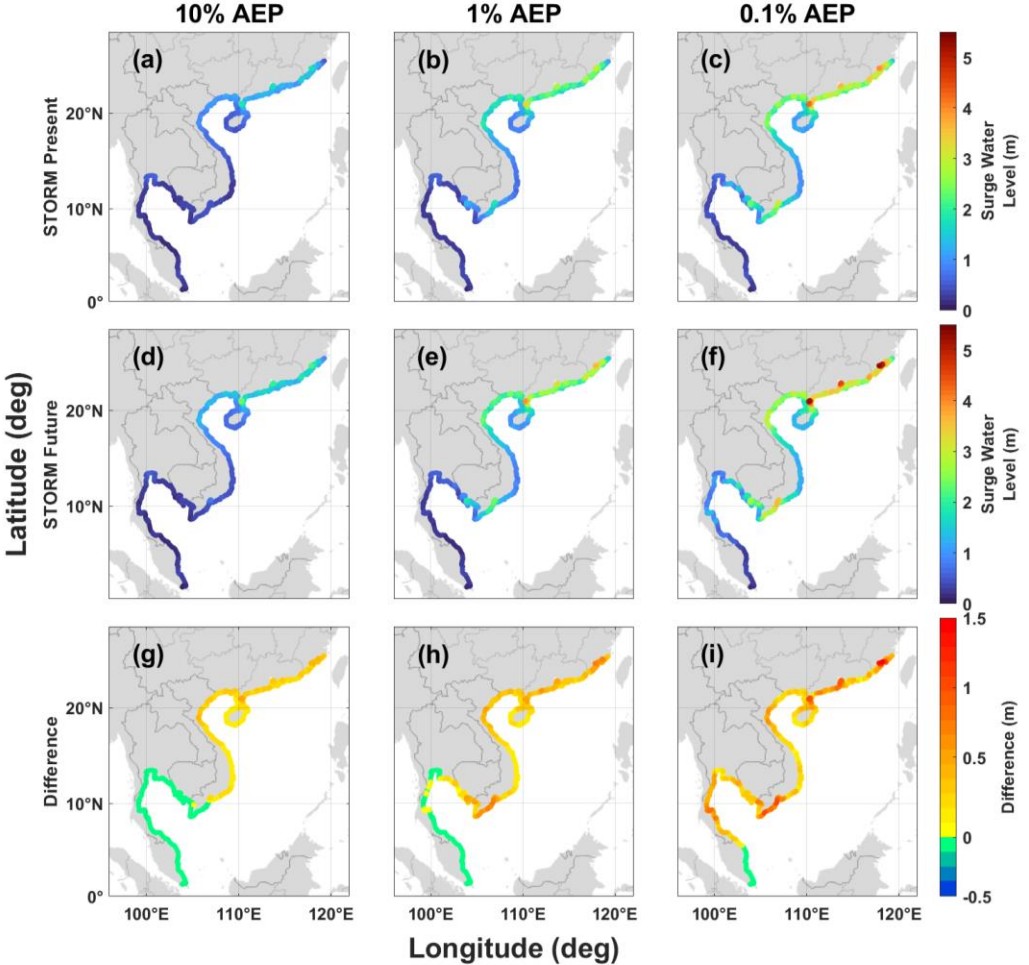

*Figure 6 - The 10% AEP (a,d,g), 1% AEP (b,e,h) and 0.1% AEP (c,f,i) storm surge heights for the China, Vietnam,*
*Cambodia, Thai and Malaysian coastlines in the model. First row: STORM baseline data (1980-2018); second row: CNRM-*
*CM6-1 climate model STORM Future data (2015-2050); third row: a difference plot to highlight the areas with greatest*
*change in surge heights between STORM baseline and STORM future model results.*

## 4.2 Extreme total water level return period levels

Second, we now focus on the total water RPLs, which combine storm surges with astronomical tides, relative to
mean sea level (MSL). The computed 10%, 1% and 0.1% total water level AEPs are shown in Figures 7a, 7b and
7c, respectively, for the coastal model grid points along the case study coastline for the baseline past scenario
(1980-2018). The spatial patterns observed along the coastline are similar to the surge only RPLs (Figures 6a, 6b
and 6c). However, total water level heights are increased, due to the addition of astronomical tides. The largest
increases (up to 2m) are observed along the southern Chinese coastline, the Gulf of Tonkin, south Vietnam, and
southern Thailand, where tidal range is largest in the study domain. Elsewhere tides add between 0.3 and 1 m.

The computed 10%, 1% and 0.1% AEPs are shown in Figures 7d, 7e and 7f, respectively, for the future scenario
(2015-2050), representing the SSP5-8.5 climate change scenario, and from the CNRM-CM6-1 climate model run.
As expected, the differences match the storm surge changes in Figures 6d, 6e and 6f, as tides are assumed to
remain constant in our approach. In southern Vietnam, the baseline 1% total water level AEP is ~ 1.9 m above
MSL on the exposed east-facing coastline, but in the future scenario this increased to 2.2 m above MSL: an

increase of 0.27 m. Similarly, in northern Vietnam, where the Red River delta is located, the baseline 1% total water level AEP is approximately 2.1 m above MSL and in the future scenario this increases to 2.4 m above MSL. The corresponding increase between the more extreme 0.1% total water levels AEP baseline and future scenario is around 0.36 m in the north, and 0.56 m in the south of Vietnam.

So far in the analysis we have just considered changes in storminess, but the area will also experience a rise in MSL due to climate change. The IPCC's 6[th] Assessment Report (Fox-Kemper et al., 2021) projects a relative MSL mean rise (relative to a 1995-2014 baseline) along the coastline of Vietnam of 0.25 m by the year 2050, under the SSP5-8.5 reference scenario (Fox-Kemper et al., 2021; NASA sea-level tool: https://sealevel.nasa.gov/ipcc-ar6-sea-level-projection-tool). The associated confidence limits of this relative MSL mean rise (at the 17[th] and 83[rd] percentile according to the IPCC methodology) are 0.17 m and 0.35 m. The computed 10%, 1% and 0.1% AEP total water level RPLs, for the future scenario (2015-2050), with an additional 0.25 m of MSL rise, are shown in Figures 7g, 7h and 7f, respectively. Adding mean sea level rise to the future 1% AEP total water level RPLs increases these levels to approximately 2.7 m above MSL for the north of Vietnam and 2.4 m above MSL for the southern part of Vietnam.

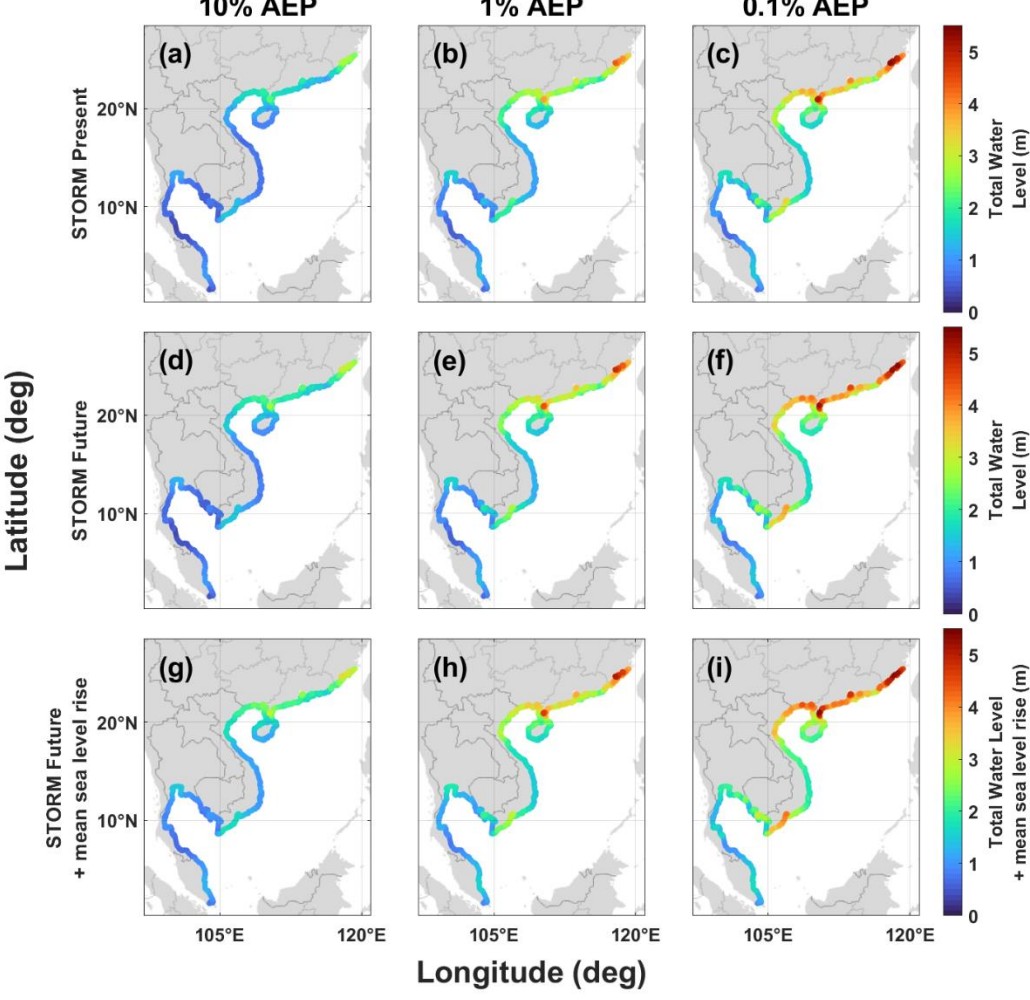

*Figure 7 - The 10% AEP (a,d,g), 1% AEP (b,e,h) and 0.1% AEP (c,f,i) total water level (tide + surge heights for the China, Vietnam, Cambodia, Thai and Malaysian coastlines in the model. First row: STORM baseline data (1980-2018); second row: CNRM-CM6-1 climate model STORM Future data (2015-2050) total water levels; third row: CNRM-CM6-1 climate model STORM Future data total water levels with 0.25m addition for rising mean sea levels up to 2050.*

To examine the results in more detail we display plots of the full range of calculated return periods in Fig.8, at 12 model grid points spaced equidistant along the Vietnam and southern China coastline, where significant and spatially varying changes were observed. The past (solid red line) and future (solid green line) total water levels are shown, along with an additional line showing the future total water RPLs with the addition of a 0.25 m of mean sea level rise (dashed blue line). In the background of all results are shaded areas showing the 95th percent confidence bounds around the mean of these total water level results. For future total water RPLs with the addition of relative mean sea level rise, these confidence bounds have 0.17 m added to the lower bound, and 0.35 m added to the upper bound, to additionally capture the uncertainty from mean sea level rise to the year 2050 using IPCC estimates, in the median SSP5-8.5 scenario, for this location. The results shown in Fig.8 highlight a few things. The first relates to coastal morphology. As mentioned above, water level RPLs are lowest around the central Vietnam coastline (points g, h in Fig.8), where the narrow continental shelf acts to reduce surge amplitude. In both the baseline and future scenarios within this central zone, the difference between the smallest (20% AEP, 1 in 5 year) and largest (0.1% AEP) total water RPL is less than ~0.8 m. This suggests that the amplitude-dampening effect provided by the coastal morphology extends to even the most severe storm surges. The second thing of note relates to changes in total water RPLs over time. Along the Vietnam coastline, the largest differences between the baseline and future scenarios occur along the exposed southern coastline. The future 1% AEP total water RPL near to the Mekong River delta (points j and k in Fig.8) is 0.6 m higher than the baseline ~1 m total water RPL value, while at the northern coastline of Vietnam near to the Red River delta (point d in Fig.8), the total water level increase over time is only around 0.3 m. The difference is only 0.12 m for the same 1% AEP return period in the central coastal of Vietnam (points g and h on Fig.8). The third thing of interest is that at most of the selected 12 grid points, the changes in future total water level return period, exceed that of a 0.25 m mean sea level rise by 2050. This highlights that under the SSP5-8.5 climate change scenario, changes in storminess are likely to dominate over changes in mean sea level rise in the coming three decades. This holds true for 1% AEP total water level, and the magnitude of the effect increases as events become more extreme (up to 0.1% AEP). The exception to this result is along the central coastline of Vietnam where surge amplitudes are consistently reduced, even at extreme probabilities, as discussed previously.

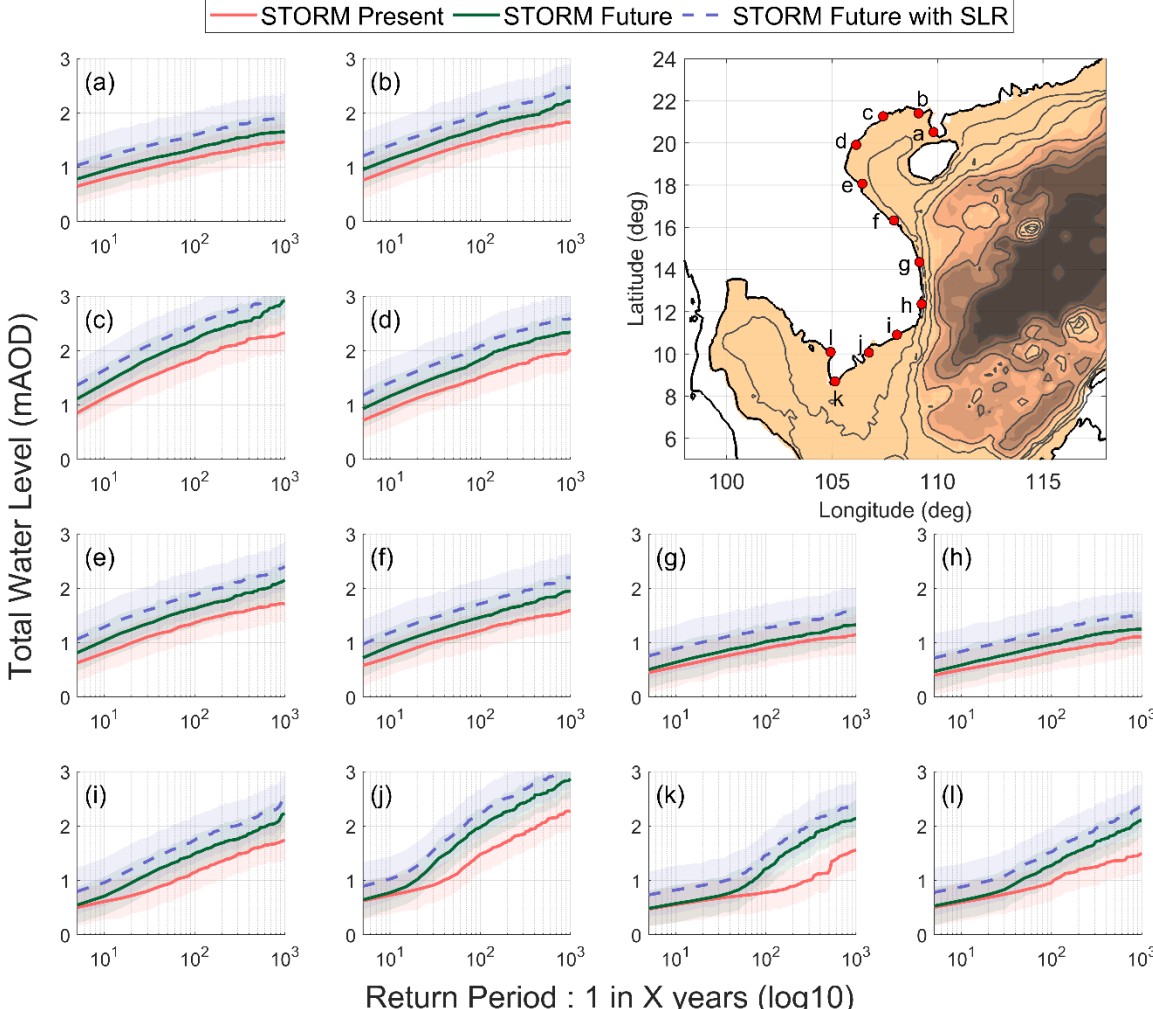

*Figure 8 - The relationship between past/present baseline (red) and future (green) total water level return period (Log scale, 1:X years) at equidistant locations at and around the Vietnam and South China coastline. Future return periods with 0.25 mean sea level rise, due to climate change by 2050, is shown with a dashed blue line. Shaded areas indicate the 95th percent confidence level around each mean total water level return period value.*

## 4.3 Cyclone tracks

Finally, we briefly examine the tracks, orientation and strength of the TC that are responsible for generating the largest storm surges in particular locations along the coastline of the case study area. Tracks of baseline synthetic TC responsible for the 10 largest modelled storm surges, located at 12 discrete points along the coastline of Vietnam and south China, are shown in Fig.9. The different TC categories are shown by different colours. There are clear differences, moving geographically north to south, in the origins and magnitudes of each TC. But what they all have in common is they pass to the south/west of each point as the TCs travel westwards within the domain. This is expected, as the Coriolis effect pushes winds in a cyclonic direction in the northern hemisphere, and it is the strong onshore winds in the first and second cyclone quadrants that are responsible for generating a large part of the storm surge. The TCs that generate the largest storm surges at the northerly points are typically associated with larger category events (3, 4 and 5), whereas for the southerly points, small category events (1 and 2) dominate.

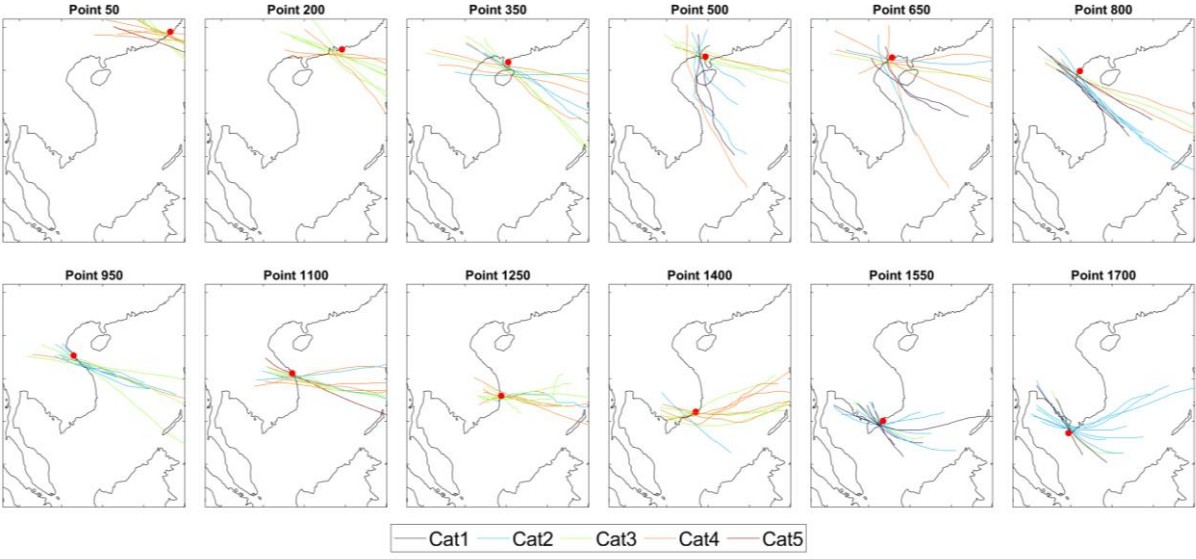

625

*Figure 9 – The tracks of the past/present STORM cyclones which produce the 10 largest surges at 12 selected coastal points along the Vietnam and China coastlines. All demonstrate that it is the onshore winds associated with the TC that are forcing storm surge levels. Before around point 1700 a counter-clockwise-turning TC travelling westwards ensures that onshore winds (and thus storm surges) are north-east of TC, and offshore winds south-west. However, after this point, with the coastline curving, onshore winds flip around so that storm surges and onshore winds are south-west of TC.*

630

## 5. Discussion

In this paper we forced a hydrodynamic coastal model of the South China Sea with wind and pressure data from a novel database of synthetic baseline and future (SSP5-8.5 - high greenhouse gas emissions) TC activity, representative of 10,000 years of TC activity in each case. Our overall goal was to gain a better understanding of the potential changes to extreme storm surge and total water level probabilities that could occur along the south China, Vietnam, Cambodia, Thailand, and Malaysia coastline under a high emission climate scenario by 2050. This area of south-east Asia is considered to be a 'hotspot' for projected future sea level extremes related to intense TC storm activity and contains many densely populated low-lying areas (McGranahan et al., 2007; Nicholls and Cazenave, 2010; Kirezci et al., 2020; Nicholls et al., 2021). Our modelling results show that a projected shift in TC behaviour under a SSP5-8.5 climate scenario, would raise surge heights along lengths of the Chinese and Vietnamese coastlines. By 2050, storm surges along the southern Chinese and Vietnamese coastlines are predicted to be to 0.8 m (1% AEP) and 1.6 m (0.1% AEP) higher than today. TC approach angle means that some north and eastern stretches of coast in the model domain would be orientated to be more vulnerable to storm surges, irrespective of their coastal morphologies, because of funnelling effects within bays and inlets (Pandey and Rao, 2019). However, coastal morphology can modulate surge heights in certain instances too. Despite storm surge heights increasing along the northern and southern Vietnamese coastline by 2050, future storm surges along the central portion of Vietnam's coastline increase much less, even though this section of coastline is most exposed to increases in the frequency and intensity of TCs that induce storm surges. Surge heights are modulated here because there is no wide and gently sloping continental shelf to amplify storm surge energy, and there are few coastal inlets and river mouths here to funnel and enhance storm surge wave heights (Jelesnianski, 1972; Dube et al., 1981).

One facet of this change suggests that a trend of TCs gradually migrating polewards and achieving their maximum intensity in more northern latitudes is expected to continue into the future (Kossin et al., 2014). This trend is in fact seen in the CNRM-CM6-1 global climate model behind the STORM data used in this analysis, along with an apparent greater number of more intense TCs occurring in the future within the WNP region. Within the smaller limited domain of our South China Sea model, we also see projected a wider distribution of activity over this region, with an increasing number of strikes to east-facing coastlines (particularly central Vietnam, Fig.5c), and even a small number of TCs travelling further southwards. The reasons were not explored but could be a seasonal effect; recent research suggests that peak season (July-September) TCs in the WNP are more likely to migrate polewards than later-season (October-December) TCs (Feng et al., 2021).

The model has a variable triangular grid resolution, with greater detail along Vietnam's coastlines. It should be noted that there is potential for sub-optimal accuracy in storm surge levels, particularly within small coastal features such as inlets, bays, or estuaries, where coastal resolution is insufficient to capture features in detail. For example, Bertin et al. (2015) showed that within small seas wave radiation can induce set up that transforms storm surge levels along exposed coastlines, with even the small waves entering bays and inlets affecting water levels. Unfortunately, the number of TC simulations entailed in this study meant that there had to be a trade-off between rendering coastal detail and reasonable computation timescales. Future work in such locations, looking at local sections of coastline, would require detailed modelling to estimate extreme water levels due to storm surge. Additionally, currents and wave action are specifically not incorporated in the modelling as it was outside of the project scope. Such wave models will be a valuable addition to the scientific discussion, as their high spatial resolution at the coastline would ensure that nearshore wave dynamics, such as wave setup, are adequately resolved (Saulter et al., 2017; Melet et al., 2018; Dodet et al., 2019; Hinkel et al., 2021).

Beyond the increased storm surge heights computed along northern Vietnam and south China, our results suggest that the effects of a changing climate on extreme sea levels will also affect more southerly latitudes around southern Vietnam, Cambodia, and parts of Thailand and Malaysia. This is troubling as currently extreme sea levels here are rare events. The 10% AEP storm surge presently averages around 0.36 m along this Vietnam-Cambodia-Thailand portion of coastline. The more extreme 1% storm surge AEP rarely exceeds 0.5 m along these coastlines, and there is so little storm surge activity that for some sections of Thailand and Malaysia coastline the difference between 10% AEP and 0.1% AEP extreme sea levels is under 10 cm. We found that the lowest-impact/highest probability storm surges (>10% AEP) along these coastlines are unlikely to greatly increase in the future. However, more extreme storm surges (1% AEP to 0.1% AEP) do increase in the future, under the SSP5-8.5 climate scenario. The worst hit sections of Vietnam-Cambodia-Thailand coastline see 1% AEP storm surge heights increase by 0.6 m (0.8 m for 0.1% AEP surges). The implications of this are that the flood defences and plans for these previously sheltered coastlines may over time become unfit for purpose, as a consequence of the projected climate changes in this region leading to TC-induced storm surges.

We also examined what happens to storm surge frequency with TCs occurring with greater intensity in the future. The gap between baseline and future % AEP suggests that the extreme storm surge levels we experience today, would occur in the future with greater regularity. For example, a 1% AEP storm surge occurring around Ho Chi Minh City (Fig.8j) with height of ~1.4 m today (excluding tide and mean sea level rise contributions), is projected to occur at close to 2.8% AEP (1 in 35 year return period) in the future, under the SSP5-8.5 climate scenario.

Storm surge levels associated with a 1% AEP event near to the Red River delta (Fig.8d) today would correspond to a 3.3% AEP (1 in 30 year return period) frequency in the future. The same effect can be observed to varying degrees for all location points plotted, with greater increase in occurrence observed in the north/south parts of Vietnam coastline (points a-f and i-l in Fig.8) than observed in the middle section of coastline (points g-h in Fig.8). This substantial increase in frequency suggests that flood defence standards will need to be upgraded at coastal locations and flood managers will need to consider augmented, alternative, or combined methodologies to cope with more widespread, higher, or more frequent storm surge scenarios. Complex and higher dyke systems alone may be insufficient for storm surge flood hazard. It's also worthwhile to consider that breaches in storm surge defences may coincide/combine with pluvial runoff or fluvial flooding after a typhoon or monsoonal rainfall when normal inland flood releases (e.g., drains, flood gates or flood storage areas) could be unavailable.

A greater number of intense TCs in the future, due to projected climate changes, also more spatially dispersed than today, means not only that extreme sea levels become higher in the future in our study area, but that the total lengths of coastline experiencing the more extreme storm surges extends also. In our analysis, the highest storm surge levels ($\geq$ 2.5 m in 1% AEP, $\geq$ 3.5 m in 0.1% AEP: approximately the 95[th] percentile of baseline coastal storm surge levels) seen today occurring only along the coastline of southern China, are projected to extend further south into Vietnam over the next 30 years. This spread would more than double the length of coastline currently impacted by such high surge levels. In Vietnam, the northern communes have more experience of TCs making landfall with some regularity and coping with TC-induced storm surges. The system of flood defences is better prepared for such eventualities. But at Vietnam's southern coastlines the population is not as well-equipped to withstand extreme sea level inundation (Kleinen 2007; Takagi et al., 2012; Larson et al., 2014; Anh et al., 2017). A significant proportion of the country's total population lives along this low-lying coastline in cities or within its two main deltas; the Red and Mekong River deltas (Dasgupta et al., 2009; Hinkel et al., 2014; GFDRR., 2015; Bangalore et al., 2019; Nicholls et al., 2021). This, alongside the considerable agricultural and infrastructure capital value, explains why these low-lying coastlines have particular vulnerability to storm surge hazard (Nguyen et al., 2007; Hung et al., 2012; Edmonds et al., 2020). The Mekong River delta has long been identified as being at particular risk of coastal flooding because mean sea levels have been historically rising here at the same time that mean land elevations have been sinking - and sinking at a faster rate than previously realised (Hung et al., 2012; Erban et al., 2014; Dang et al., 2018; Minderhoud et al., 2017; GSO, 2019; Oppenheimer et al., 2019; Nicholls et al., 2021). As one of the most strategically important areas of the entire model domain, we aim to explore the potential impacts of predicted extreme total water levels to the Mekong River delta region in a future paper.

Mean sea level rise has not been explicitly incorporated in our future model simulations, but it would likely have an amplifying effect on extreme sea levels. To accommodate the impact of mean sea level rise for the Vietnam coastline as an example, we instead simply added 0.25 m increase on top of total water level results (Fig.8). We find that the projected increases in storm surge heights along the entire Vietnamese coastline in the future, under the SSP5-8.5 climate scenario, can be up to 0.6 m in the 1% AEP measure (and 0.8 m in the more extreme 0.1% AEP). Surge heights are even higher along south China coastlines (Fig.6h). But even an average of all 1% AEP surge level increases along this coastline actually exceeds the anticipated permanent addition due to climate change on local MSL. Consequently, storm surge would appear to present a bigger (albeit limited time) hazard to this region than rising MSL, by 2050. This is particularly interesting, as past changes in MSL have dominated

changes in extreme sea levels in extra-tropical regions, with mostly negligible changes observed in storm surges (Seneviratne et al., 2012; Marcos et al., 2015; Mawdsley and Haigh 2016). The results of our study, which highlights the large - and growing - impacts of storm surge flooding, clearly demand an urgent re-evaluation of existing flood risk, defence design and planning standards to include appropriate focus on the emerging risks posed by climate driven storm surges. High value areas south of around 15 degrees latitude which have historically disregarded the risks posed by current and future storm surges, have the strongest exposure to this risk (Takagi et al., 2012; Anh et al., 2017).

There are factors that influence extreme sea levels (such as wave run up and set up, TC latitude or seasonality of MSL) that have not been incorporated in our model set up as they are currently beyond the scope of the project. However, they could easily be incorporated in future analysis. For example, we constructed TCs for the MIKE 21 FM model using the Holland method (Harper and Holland, 1999), but alternative approaches may produce slightly different TC wind and pressure gradients in the model to induce storm surge heights. Naturally, there are also alternative choices that could have been made in our study approach that would or may have altered our findings. For example, we selected a single future STORM scenario (CNRM-CM6-1) out of a possible four climate model outputs and any biases in this data would also translate into our model results. However, all STORM versions of the averaged 2015-2050 future climate (Table 3) consistently showed an increase in TC intensity, frequency and altered spatial distribution in the South China Sea region. Future work could compare results across the three other climate simulations to better quantify uncertainty. We also highlight again that waves, particularly wave set up and run up, are important contributors to extreme sea level and coastal flooding, particularly in areas of intense TC activity. Due to project time, computation outlay, and limit of budget that funded this study, we have focused in this paper only on storm surges and still sea levels. Future work could include waves. The same framework could be applied to simulate past/present and future wave climates and incorporate them in estimates of total water level probabilities.

Finally, we stress that in this paper we have utilised the future STORM database from Bloemendaal et al. (2022) that is based on a SSP5-8.5 climate change scenario. Note, Bloemendaal et al. (2022) only created TC for the SSP5-8.5 scenario and no others, so we were not able to run other climate projections. SSP5-8.5 assumes a future society that has developed within the highest greenhouse gas emissions pathway, being more energy-consumptive but also successfully using innovation and technology to adapt, rather than mitigate, against its environmental problems. It has less social and economic inequality compared to most other pathways and has a booming global economy. A SSP5-8.5 future represents a low-likelihood outcome for 2100 (Hausfather and Peters 2020; IPPC 2022; Pielke et al., 2022) and that the most plausible scenario for 2100 is thought to be closer to SSP2-4.5 and SSP3-7 if pledges and global climate progress so far are incorporated. Nevertheless, the carbon gap between what we have now and what we ought to achieve by 2050 looms large (Hausfather and Peters 2020; Pielke et al., 2022). It will require an enormous global effort to achieve the policy goal of global net-zero $CO_2$ emissions by 2050, not least because this policy relies on decarbonisation technologies for decades to come (IEA, 2022; Pielke et al., 2022). Climate projection outcomes to 2100 are provisional, which means our climate by 2050 is unknown. This uncertainty is also seen in the International Energy Agency's (IEA) research which suggests that populations in 2050 would be living through an intermediate era, with $CO_2$ emission levels that have plateaued before levels dip further by 2100 (Lee et al., 2021; IEA, 2022; Pielke et al., 2022). Indeed, Schwalm et al. (2020) argues that the SSP5-8.5 scenario is, in fact, the best tool to quantify physical climate risk by 2050, because this scenario has so

far most closely tracked the total cumulative $CO_2$ emissions to date. Nevertheless, whether SSP5-8.5 represents the future worst case in global emissions by mid-century, or whether it truly characterises the mid-point of global climate on the path to best-case outcomes, SSP5-8.5 outputs have real value when attempting to define future storm surge flood risk response. We therefore believe the model outcomes in this paper provide useful data not only for decision-makers tasked with developing flood policy to serve future generations, but also for sectors thinking how to develop flood defences that age well because they are capable of withstanding the worst-case-scenario storm surges of the future.

## 6. Conclusions

As the latest IPPC report has indicated (Fox-Kemper et al., 2021), there is currently little (~20%) confidence in the scientific community being able to accurately predict future changes to storm surge characteristics, particularly in regions of the world exposed to TCs. The low level of confidence arises both because of the significant challenge of predicting changes in TC activity at a local and regional scale, and because relatively few studies have assessed changes in storm surge driven by TCs. Therefore, our overall aim in this paper is to apply a novel modelling framework to more accurately estimate both present and future storm surge and extreme sea level hazards, by considering the densely populated coastlines of south China, Vietnam, Cambodia, Thailand, and Malaysia as a case study.

We configured a depth averaged hydrodynamic model of the South China Sea and extensively validate it against measured sea level data from tide gauges in the region. We then forced the hydrodynamic model with 10,000 years of TC activity, representative of a past (1980-2017) and future (2015-2050) future period, based on a high-emission climate projection scenario. From the model outputs, we estimate both past and future storm surge and extreme sea level probabilities along the coastlines of south China, Vietnam, Cambodia, Thailand, and Malaysia.

Our results showed that extreme storm surges, and therefore total water levels, increase substantially in the coming decades (up to 2050), under a high emission (SSP5-8.5) scenario, driven by an increase in the frequency of intense TCs. The increases in storm surges in some regions, e.g., along the south China, and northern and southern Vietnam coastlines, can exceed ~1 m in the 1% AEP measure; significantly more than the expected changes in mean sea level rise over this period. The length of coastline that is currently exposed to storm surge levels of 2.5 m or greater more than doubles (353 km to 930 km), between the baseline and future high emission scenario. Around the low-lying and densely populated areas of the Red River and Mekong Delta storm surges with an AEP of 1% (1 in 100 year return period) today, are likely to see a change in frequency to ~3% AEP (1 in 30 year return period) over the coming decades. Furthermore, at higher return periods, the coastlines of Cambodia, and parts of Thailand and Malaysia are predicted to experience storm surges induced by TCs in the future scenario, whereas presently they don't. A similar methodology to that applied here could be used to assess changes in storm surges and extreme water levels in other regions of the world that are exposed to TC activity.

Many future projections of extreme sea level, at global, regional, or local scales, only account for changes in relative mean sea level, but here we have shown that changes in storm surges could be significant, and even exceed changes in MSL in some areas. Our study area has many low-lying and densely populated coastlines, such as the major river deltas in this region, that are especially vulnerable to storm surges. Given these findings, coastal flood

management, planning and adaptation in these areas should be reviewed for their resilience against changes in storm surges and total water level levels in the future.

**Code/Data availability**

The past/present and future 10%, 1% and 0.1% AEP return period storm surge and total water levels for the modelled coastline of south China, Vietnam, Cambodia and parts of Thailand, as described in section 4.1 and illustrated in Fig 6, are provided for reference in the National Oceanography Centre British Oceanographic Data Centre Published Data Library:

https://www.bodc.ac.uk/data/published_data_library/catalogue/10.5285/e17e7db6-4a78-1a89-e053-

6c86abc0253d/

## Author contribution

MW carried out the hydrodynamic modelling simulations, formal analysis and prepared the first draft of the manuscript. IDH, SED and RJN conceptualised this work, designed the methodology and provided supervision in the UK. IDH further supervised model design and visualization. The supervision, methodology development and resources in Vietnam were provided by TBH and NNH, with formal data analysis carried out by QQL. NB provided key TC resources for this study. All authors commented and edited the manuscript prior to submission.

## Competing interests

The authors declare that they have no conflict of interest.

## Acknowledgements

This work was supported by the Natural Environment Research Council, in the UK (Grant Number NE/S003150/1). Also supported in Vietnam by the National Foundation of Science and Technology Development (NAFOSTED-RCUK fund) and with data from the Ministry of Science and Technology of Vietnam's river training solutions of the Mekong River project (code DTDL-48/18). The authors also wish to acknowledge the use of the IRIDIS High Performance Computing facility, and associated support services at the University of Southampton, in the completion of this work. Lastly, this work could also not have been completed without access to the STORM present and future datasets (Bloemendaal et al., 2022). NB was funded by a VICI grant from the Netherlands Organization for Scientific Research 569 (NWO Grant Number 453-13-006) and the ERC Advanced Grant (Grant Number COASTMOVE 884442).

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

**Appendix 1**

**Section 1: Flow charts and tables to illustrate model configuration, validation, and simulations**

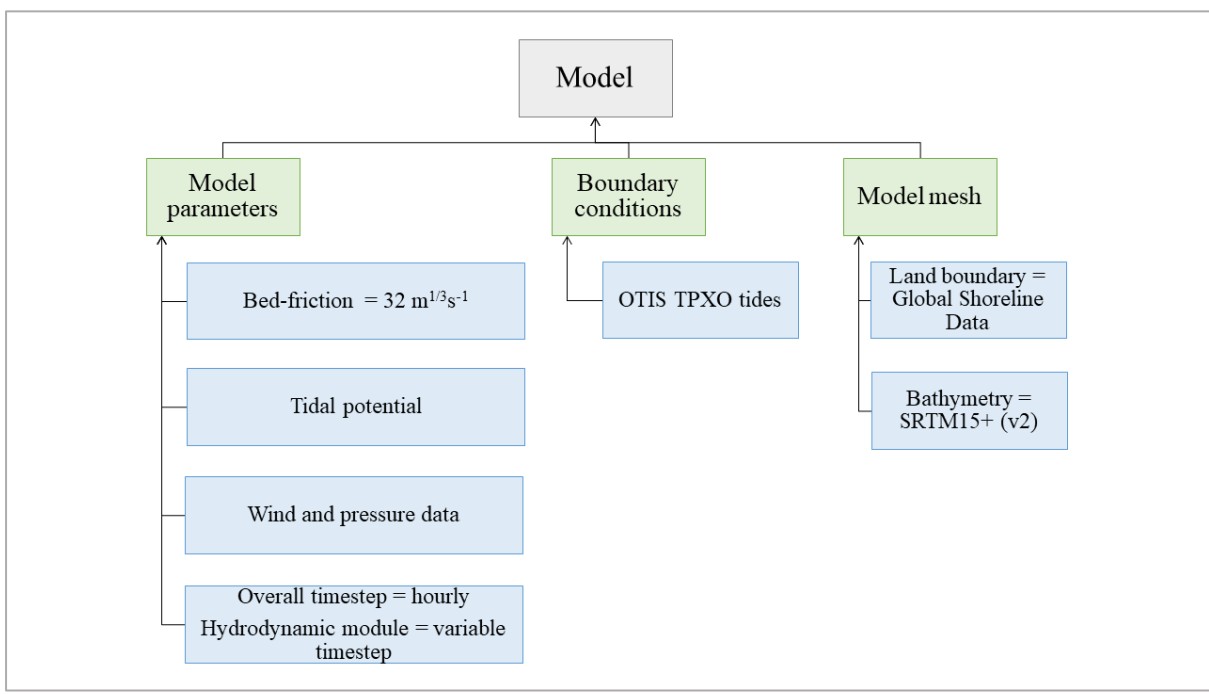

*Figure A1 – The basic model configuration*

*Table A1 – MIKE 21 FM set up variable and values.*

| Parameter | Sub-parameter | Description |
|---|---|---|
| **Time** | - | Number of timesteps = 5543<br>Timestep interval = 3,600 seconds |
| **Hydrodynamic Module** | **Solution Technique (for both the shallow water equations and transport equations)** | Time integration = Higher order<br>Space discretization = Higher order<br>Minimum time step = 0.01 seconds<br>Maximum time step = 25 seconds<br>Critical CFL number = 0.8 |
| | **Depth** | No depth correction |
| | **Flood and Dry** | Type = flood and dry<br>Drying depth = 0.01 m<br>Wetting depth = 0.1 m |
| | **Density** | Density type = barotropic (default) |
| | **Eddy Viscosity** | Eddy Type = Smagorinsky formulation (default values) |
| | **Bed Resistance** | Resistance Type = Manning number (default) – whole domain<br>Manning number data:<br>Format = Constant (default)<br>Constant Value = 32 $m^{1/3}$ $s^{-1}$ (default) |
| | **Coriolis Forcing** | Coriolis Type = varying in domain (default) |
| | **Wind Forcing** | (files created as described in manuscript) |

| Parameter | Sub-parameter | Description |
|---|---|---|
| | Ice Coverage | No ice coverage (default) |
| | Tidal Potential | Yes, to include Tidal Potential (default values used) |
| | Precipitation - Evaporation | No precipitation (default) |
| | | No Evaporation (default) |
| | Infiltration | No infiltration (default) |
| | Wave Radiation | No wave radiation (default) |
| | Sources | No change (default) |
| | Structures | No change (default) |
| | Initial Conditions | Constant (default) |
| | | Initial data surface elevation = 0 m (default) |
| | | u-velocity = 0 ms$^{-1}$ (default). v-velocity = 0 ms$^{-1}$ (default). |
| | Boundary Conditions | 7 tidal boundary conditions. Files created as described in main manuscript. |
| | Decoupling | Do not include (default) |
| | Outputs | Aerial and point data as described in the manuscript. |

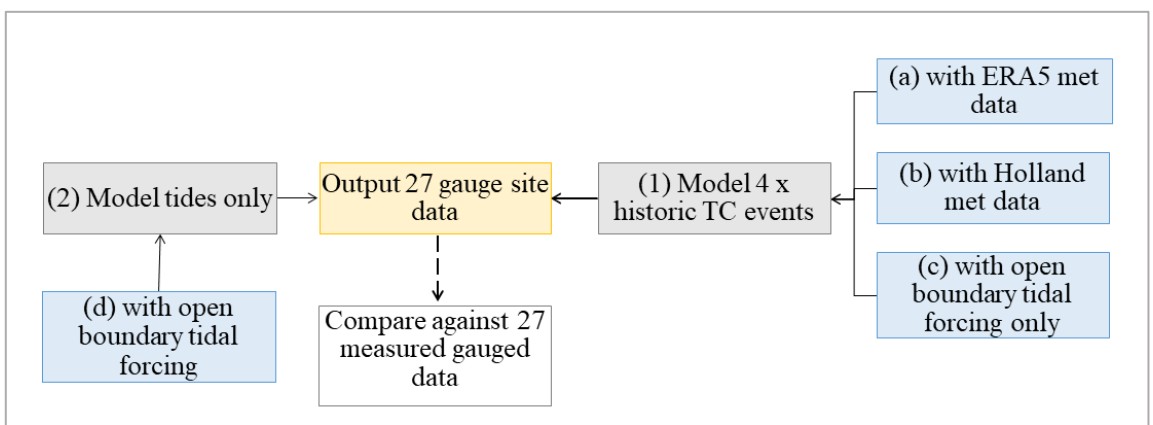

*Figure A2 – Model validation. The storm surge validation process is described in section 2.3 of the paper; we use different wind and pressure input data to simulate historic TC events, using (1,a) ERA5; (1,b) Holland formula or (1,c) no meteorological forcing data. Separately, validation of astronomical tides is illustrated in schematic (2,d) with a tides-only model simulation, as described in Section 3 below, and in section 2.2 of the paper.*

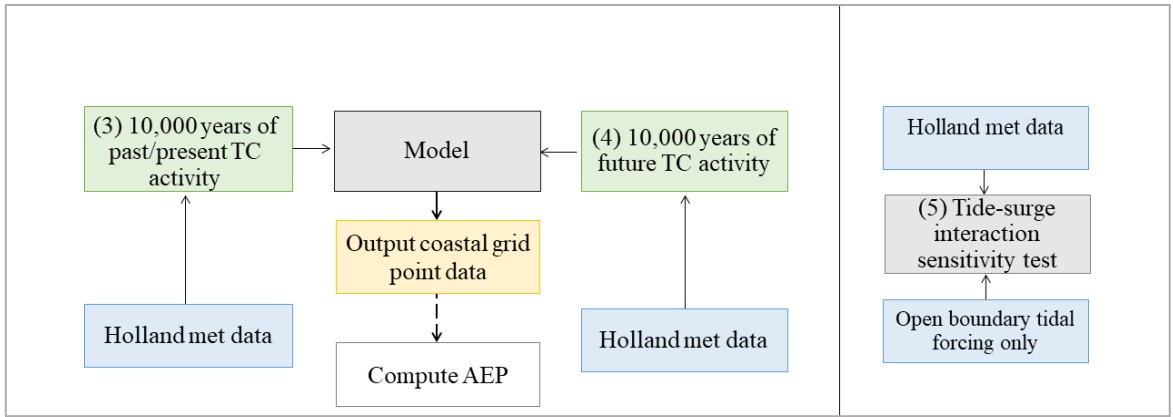

*Figure A3 – MIKE 21 hydrodynamic model configuration for (3) past/present scenario; the (4) future scenario and (5) testing the model sensitivity to tide-surge interactions (see Section 3).*

## Section 2: Storm surge model validation

Figures S4-S6 below show the simulated storm surges achieved using this approach for Typhoons Sally (September 1996), Tropical Storm Linda (November 1997) which passed over the southern tip of Vietnam and Mangkhut (September 2018). These results contrast ERA5 (red dashed) and IBTrACS (red dotted) data, at the node point nearest the measured tide gauge location, against measured data (blue). A green vertical line indicates the date/time of nearest landfall of the TC. Both (a) total water level and (b) surge-only water levels are shown.

Table A1 additionally provides the root mean square difference between measured and modelled total water levels and surge sea levels from these validation simulations (calculated for the number of days shown in Figures S4-S6).

*Table A2 - The Mean Absolute Error (m) between (a) measured tide gauge and modelled total water levels and (b) tide-removed measured data and modelled surge-only water levels from all the validation hindcast simulations.*

| Typhoon name (date) | Total water level mean absolute error (m) | | Storm surge level mean absolute error (m) | |
|---|---|---|---|---|
| | ERA5 data | Holland model | ERA5 data | Holland model |
| Sally (September 1996) | 0.24 | 0.21 | 0.17 | 0.11 |
| Linda (November 1997) | 0.32 | 0.21 | 0.26 | 0.17 |
| Mangkhut (September 2018) | 0.25 | 0.29 | 0.19 | 0.11 |
| Ketsana (September 2009) | 0.31 | 0.15 | 0.24 | 0.08 |

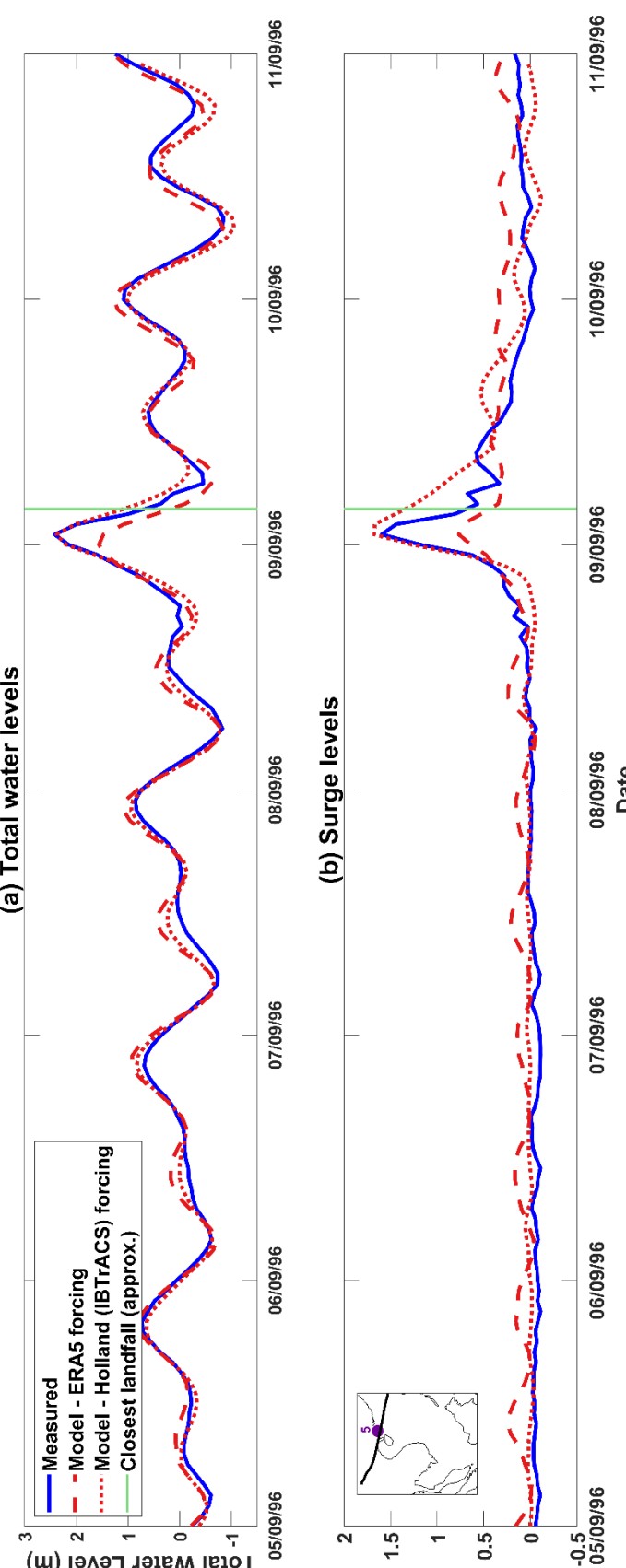

*Figure A4 - (a) measured and modelled total water levels, and (b) measured and modelled storm surges at tide gauge 5: Zhapo, China (inset or see Fig. 1 for location), for Typhoon Sally, which made landfall on 9th September 1996 (green vertical line). Modelled total water level and surges using ERA5 (red dashed) and Holland Model (red dotted) wind and pressure fields against measured data (blue).*

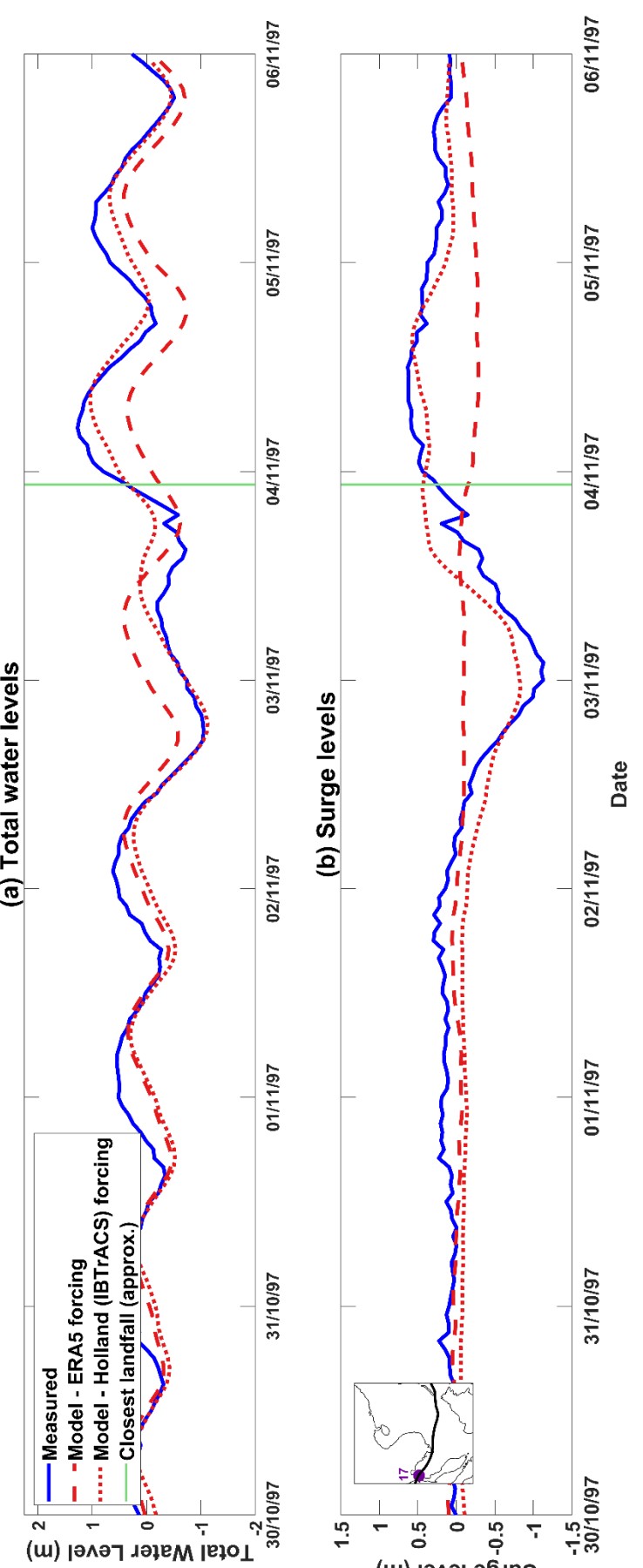

*Figure A5 - (a) measured and modelled total water levels, and (b) measured and modelled storm surges at tide gauge 17: Ko Lak, Thailand (inset or see Fig. 1 for location), for tropical storm Linda, which made landfall late on 3rd November 1997 (green vertical line). Modelled total water level and surges using ERA5 (red dashed) and Holland Model (red dotted) wind and pressure fields against measured data (blue).*

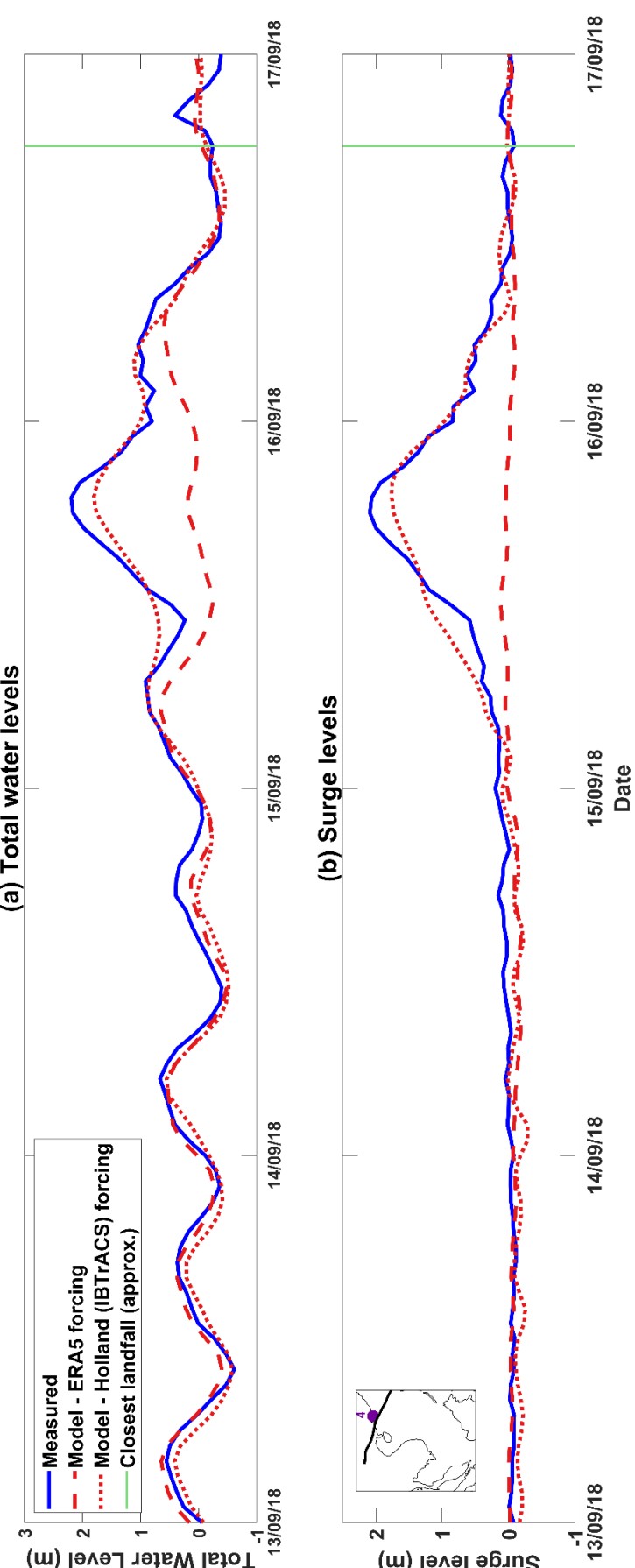

*Figure A6 - (a) measured and modelled total water levels, and (b) measured and modelled storm surges at tide gauge 4: Hong Kong, China (inset or see Fig. 1 for location), for Typhoon Mangkhut, which made landfall in the evening of 16th September 2018 (green vertical line). Modelled total water level and surges using ERA5 (red dashed) and Holland Model (red dotted) wind and pressure fields against measured data (blue).*

**Section 3: Tide-surge non-linear interactions sensitivity tests**

We undertook sensitivity tests to assess whether it was necessary to include the astronomical tide when simulating sea level conditions for each cyclone. Simulating tides adds the complexity that each cyclone must be randomly assigned a specific day and start time, but STORM only assigns each cyclone to a year and month. Customarily extreme sea level modelling studies simulate storm surges separately from the tides and then statistically combine the two to estimate total water level return periods (e.g., Dullaart et al., 2021). However, this approach ignores the fact that non-linear interactions between the tide and non-tidal components of sea level have been reported for many places around the world and typically result in the highest observed non-tidal residuals occurring around mid-tide or low-tide rather than at the time of tidal high water (e.g., Horsburgh and Wilson, 2007; Rego and Li 2010; Mawdsley and Haigh, 2016; Williams et al., 2016; Poulose et al., 2018; Idier et al., 2019). Arns et al. (2020) have recently shown that flood risk can be underestimated if these non-linear interactions are not accounted for, as Haigh et al. (2014) discovered with storm surge errors exceeding 1m along the Australian coastline, when surges were simulated independently of tide for Category 5 Tropical Cyclone Rosita (2000).

In our approach to assess the significance of non-linear interactions in this region, the first step was to create model results with meteorological forcing only. We recreated Typhoon Ketsana (2009) using data from the IBTrACS database, and the MIKE 21 FM Cyclone Wind Generation tool (DHI, 2017b) with Holland B parameter estimated using the Holland Formula (Harper and Holland, 1999). Then, four comparative model runs were undertaken in which the model was driven with both tidal and meteorological forcing. In these four comparative simulations we shifted the timing of the meteorological forcing so that the peak of the surge occurred: (1) around the time of low tide; (2) on the rising tide; (3) at high tide; and (4) on the ebb tide. The results are shown in Fig. A7 and highlight that differences in the height and timing of the simulated surge between tidal and meteorological forcing versus meteorological forcing alone, around the mid-section of Vietnam, are nominal. Whilst the maximum height and duration of the event is not affected, the shape and timing of the surge peak is impacted by as much as 0.25 m / 6 hrs between high and low tidal states. This influence should be acknowledged. But because this difference is small relative to the Vietnam tidal range, for simplicity in our study we implemented all hydrodynamic model simulations as meteorological forcing only ('surge-only') with the intension that results may afterwards be added to a randomly selected maximum tide to compute total water levels and associated return periods.

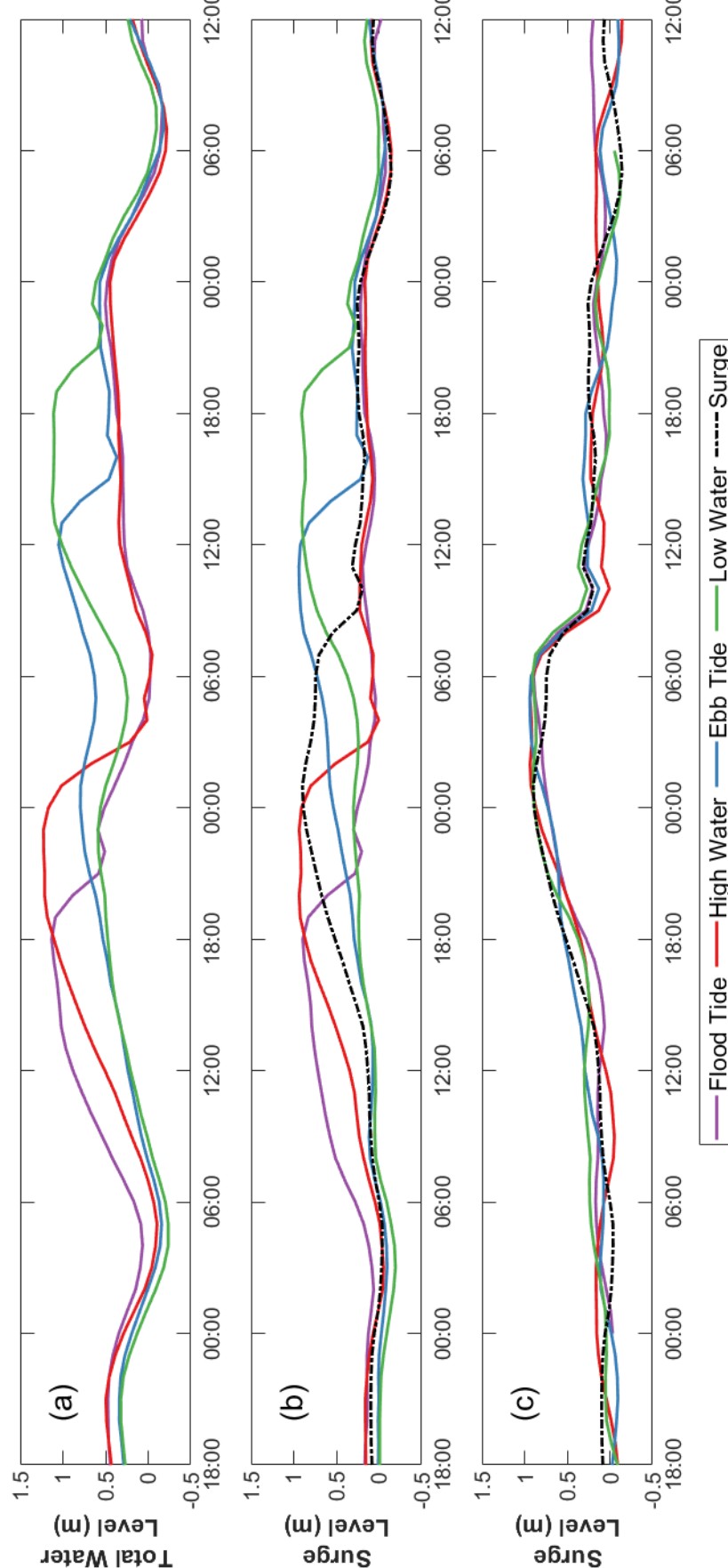

*Figure A7 - Tide-surge interaction for the Saffir Simpson scale category 2 Typhoon Ketsana at Son Tra, Vietnam (11 in Fig. 1) which made landfall on 29 September 2009: (a) total water levels for low to high tide states; (b) tide states surge-only levels and (c) tide states surge-only levels adjusted for time offset from peak.*

**Section 4: Track density for three alternative climate models to the CNRM-CM6-1 used in this analysis**

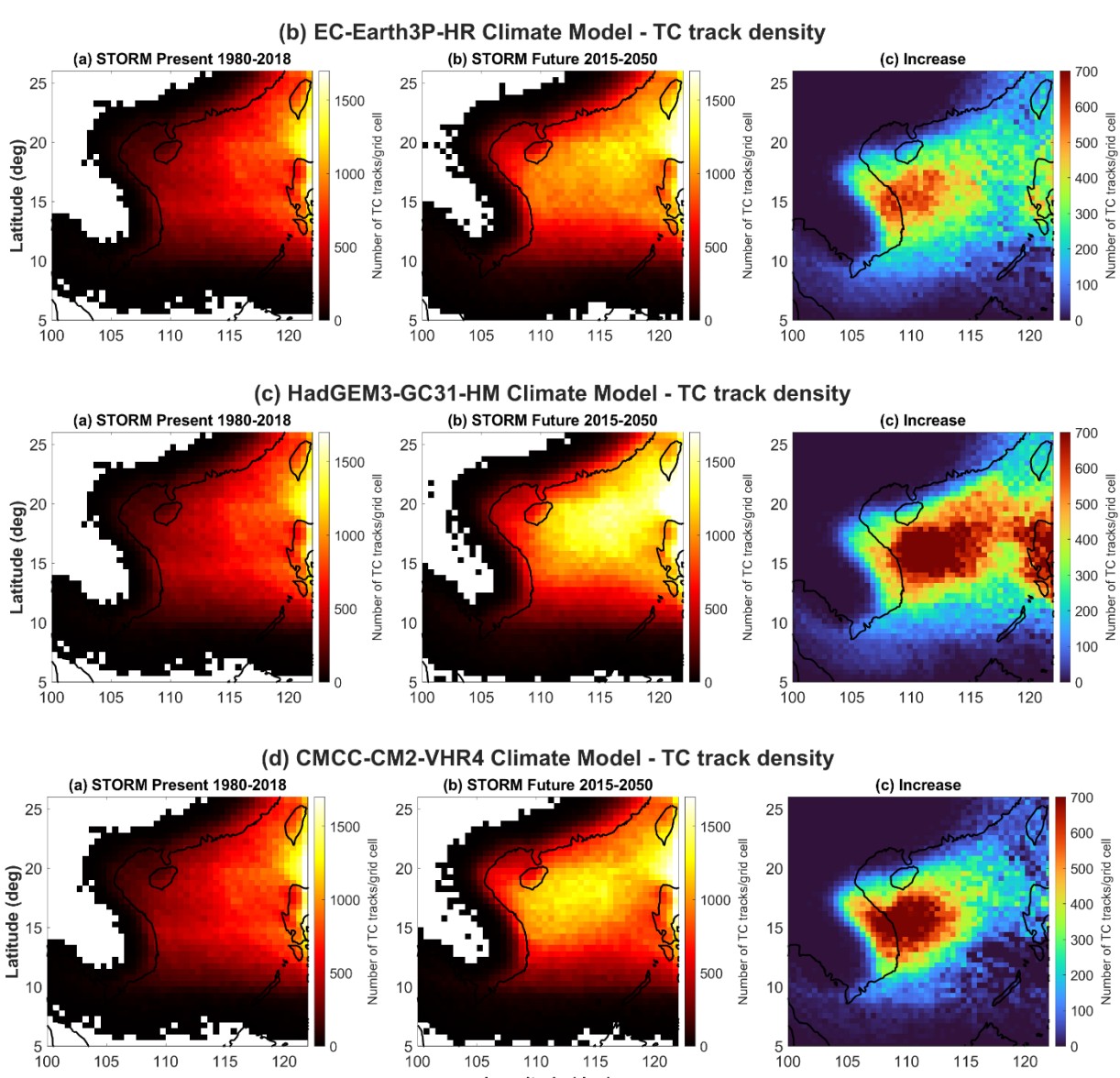

*Figure A8 – To contrast with Figure 5 in the manuscript, here are the number of Saffir Simpson Category 1+ (i.e., excluding Tropical Storms) TCs that pass through a grid cell over a ~35 year period represented by STORM Past/Present (1980-2018, left column); and STORM Future (2015-2050, central column) for three alternative climate models. Right column: the difference between STORM past/present and STORM future (i.e. the projected increase). (b) EC-Earth 3P-HR climate model, (c) HadGEM3-GC31-HM climate model, (d) CMCC-CM2-VHR4 climate model.*