# Peer review of "Climate-induced storminess forces major increases in future storm surge hazard in the South China Sea region"

_Natural Hazards and Earth System Sciences, 2021_

## Author Comment (AC1)

[Figure]

*Fig S1 – New - Validating modelled surges using ERA5 (red dashed) and Holland Model using IBTrACS (red dotted) wind and pressure fields against measured data (blue): Typhoon Sally surge at tide gauge 5: Zhapo, China (inset or see Figure 1 for location), located closest to Zhapo station in the early hours of 9th September 1996 (green vertical line). Firstly (a) comparing total sea levels, and then (b) comparing surge-only water levels.*

*Table S1 – New - The Root Mean Square Error (m) between (a) measured (tide gauge) and modelled total water levels and (b) tide-removed measured data and modelled surge-only water levels from all the validation simulations.*

| Typhoon name (date) | (a) Total Water Level RMSE (cm) | | (b) Surge-only Water Level RMSE (cm) | |
| --- | --- | --- | --- | --- |
| | ERA5 data | IBTrACS data + Holland model | ERA5 data | IBTrACS data + Holland model |
| Maring (September 1996) | 33 | 30 | 20 | 15 |
| Linda (November 1997) | 41 | 28 | 39 | 20 |
| Mangkhut (September 2018) | 42 | 20 | 36 | 15 |
| Ketsana (September 2009) | 40 | 18 | 34 | 10 |

[Figure]

*Fig 5. Revised - Left: Baseline STORM track density of Saffir Simpson Category 1+, Middle: CNRM climate model- Future STORM track density, of Saffir Simpson Category 1+, and Right: The cyclone path density difference between them (Saffir Simpson Category 1+ only - i.e. excluding Tropical Storms). Track density is the number of tracks passing through a grid cell.*

---

## Author Comment (AC2)

[Figure]

*Figure S1 - Validating modelled surges using ERA5 (red dashed) and Holland Model using IBTrACS (red dotted) wind and pressure fields against measured data (blue): Typhoon Sally surge at tide gauge 5: Zhapo, China (inset or see Figure 1 for location), located closest to Zhapo station in the early hours of 9th September 1996 (green vertical line). Firstly (a) comparing total sea levels, and then (b) comparing surge-only water levels.*

[Figure]

*Figure S2 - Validating modelled surges using ERA5 (red dashed) and Holland Model using IBTrACS (red dotted) wind and pressure fields against measured data (blue): tropical storm Linda storm surge at tide gauge 17: Ko Lak, Thailand (inset or see Figure 1 for location) which made landfall late on 3rd November 1997 (green vertical line). Firstly (a) comparing total sea levels, and then (b) comparing surge-only water levels.*

[Figure]

*Figure S3 - Validating modelled surges using ERA5 (red dashed) and Holland Model using IBTrACS (red dotted) wind and pressure fields against measured data (blue): Typhoon Mangkhut storm surge at tide gauge 4: Hong Kong, China (inset or see Figure 1 for location). Firstly (a) comparing total sea levels, and then (b) comparing surge-only water levels. Mangkhut made landfall west of Hong Kong in the evening of 16th September 2018 (green vertical line).*

[Figure]

*Comparison of Figure 2b- Left is original, Right is updated bathymetry and colorbar. [ this is SRTM15+ ocean bathymetry for nearshore in our South China Sea domain, showing down to 250 m depth only, and the specific Output Coastline Points (black) exported from the model.]*

---

## Author Response (AR1)

Thank you for the thoughtful comments from each of the two reviewers. We have carefully considered each comment and below outline the changes we have made. We believe these changes have strengthened the manuscript and so thank each reviewer for their time.
* * *
**REVIEWER #1 COMMENTS**

**1.      Figure 1: change "model location, with location of" to "model domain, with locations of". Also, I think there is a typo "at ~ 250 km depth", it should be "at ~250 m depth".**

A1. Thank you, we have made this change.

**2.      Table 1 should be moved to Section 2.1 for discussion. Also the title of Table 1 indicates values reported are for tidal constitutes, which constitutes?**

A2. Thank you for spotting this discrepancy in the caption; we have now corrected it to read '*Validation of sea levels output by the model'.* We did not in the end move Table 1 to Section 2 (The hydrodynamic model), but instead inserted it into the newly created 'Data and Methods' section, where it is now first mentioned.

**3.      I have some concerns on the value reported in the Table, it is very questionable that the "mean tidal range" is greater than 8.5 m for Xiamen, 7 m for Beihai, 5.9 m for Vong Tau. Figure 3 clearly showed that the max tidal range at Vong Tau is less than 4 m. Please check and clarify.**

A3. We thank the reviewer for pointing out this inconsistency and we shall examine these calculations more closely. We decided however, for this paper, to remove this column entirely from the table as it wasn't strictly adding a great deal towards the core message.

**4.      The model resolution of 11 km along mostly the coastal, with finest of ~2.3 km along the coastline of Vietnam, is not sufficient to represent the details of the shoreline and bathymetric feathers, which are critical for storm surge simulations. Discussion on model accuracy in estuaries and bays is necessary to clarify that the model results should be treated with care in those areas.**

A4. Thank you for your comments and we understand your concerns. We do agree this model resolution is not adequate to represent storm surge detail around estuaries and bays and as such have added text to the discussion section to acknowledge that this resolution does not capture the level of detail in these areas. The discussion section now states, '*It should be noted that there is potential for sub-optimal accuracy in storm surge levels, particularly within small coastal features such as inlets, bays or estuaries, where coastal resolution is insufficient to capture features in detail. For example, Bertin et al. (2015) showed that within small seas wave radiation can induce set up that transforms storm surge levels along exposed coastlines, with even the small waves entering bays and inlets affecting water levels*.'

There were a couple of reasons why we did not model at higher resolution along these shorelines. Firstly, we wanted to simulate 10,000 years of Tropical Cyclone (TC) behaviour, and to do this twice-over (once for the present day and once for the future scenarios). Each synthetic year normally also having multiple TC events, within our region of interest. The execution of all these model simulations would be both computationally expensive and time-consuming to get through. Therefore, we aimed to strike a balance between model resolution and duration run time. Given our regional focus, we felt 2.3 km was a good compromise that enabled us to capture as much of the TC information as possible. In the event, even with use of a High Performance Computer at the university, we spent around ~6 months to just run all the various models.

There is also the fact that this project was supported by UK and Vietnam Government funding. Our Vietnam colleagues do not have access to a High Performance Computer, and they wanted to be able to use the model for other applications. Hence, our second reason was our aim to keep the model resolution at a point that our project partners could run it on a standard desktop computer after the project ends.

**5.        Section 2.3 and Figure 4: storm surge validation is important for this study. It is not sufficient to use Figure 4 to demonstrate the Holland Model is better than ERA5. Model-data comparison at more stations and error statistics should be provided.**

A5. A good point and thank you for highlighting this omission. We used TC Ketsana as an example because the event was really nicely captured as it passed fairly close to a tide gauge. As discussed within the paper a common problem is that TC events are difficult to capture on record, due to proximity or timing. However, a small handful of historic TC events were similarly close enough to a recording device to be used to determine the best methodology and validate model, and these additional TC events are now shown within the supplementary section. One such TC event is Typhoon Sally (aka Typhoon Maring) which struck Zhapo, China in September 1996 the figure of which shown in the reproduced in the below Amended Figures and Tables section (Figure S1).

A table showing the validation RMSE between observed and modelled results, for these total four historic TC events, is also now provided within the supplementary section (also copied in the attached Figures and Tables document). Furthermore, the text of the main paper has been updated to replace 'not shown' in Line 263 with '*shown in the supplementary section*', to point to this new information.

**6.        Figure 5. Are the values shown in the figure the number of cyclones? Please provide unit in the legend.**

A6. We are happy to update the legend of Figure 5 to make it clearer. The below Amended Figures and Tables section shows this update. Yes the values are the number of cyclone tracks passing through each gridded location, within the ~35 year assessment period. Figure 5c shows a heatmap of where activity is estimated to increase most intensely over time, under the CNRM 2015-2050 climate projection.

**7.        Line 449: statement "…that tides intensify storm surge hazard" implies the nonlinear effect of tide and surge interaction but in ~Line 336 the authors showed the nonlinear effect is negligible and therefore the total water level can be calculated by the sum of two separate runs: tide and storm surge. Please clarify.**

A7. This is now corrected and reworded in Lines 449 and 499 - thank you for finding this inconsistency in the document. Using 'tides intensify storm surge hazard' was a poor choice of words since we proved that the nonlinear effect is negligible along the shorelines of interest.

**8.        How tidal elevation is added to storm surge to obtain the total water level since the synthetic TC does not have a realistic time and date?**

A8. Thank you for your comment. It is correct to say that the database of synthetically generated TCs does not assign literal dates. However, there is a synthetic year (between year 1 and year 10,000) and a month associated with each generation. Our approach was therefore to (1) generate tide levels for the model domain, covering a recent 19-year period, to encompass the full 18.6 year nodal cycle. Then (2) assign each synthetic TC a new random year, day and time, but leaving it with the same month as generated. Then (3) combine the modelled storm surge and tide together taking care that the calendar month of the TC and tide were the same - ensuring the storm surge occurs (synthetically) in the correctly aligned season/time of year. This 19-year reproduction of tides and storm surges was afterwards used to generate statistics for total return period levels that are illustrated in Figure 9.

We have clarified this in the final paragraphs of Section 2.2.3 of the paper which now reads, '... *we also estimated total still sea level return periods for each OCP. Because non-linear interactions between tide and non-tidal components were determined to not be an issue for this region, we do this by adding surge levels to a semi-randomly selected tide. The first step to calculate total still sea levels was to run a tide-only model simulation (for the year 2009 where we had already obtained OTIS tide data) and to save modelled tide levels at each of the OCPs at 10-minute intervals. We then secondly input this year of detailed tide levels into the MATLAB T-Tide script (Pawlowicz et al., 2002) to obtain the tidal constituents and predict the tides over a longer recent 19-year period (2003 to 2021). A full 19-year period was targeted because it encompasses a complete 8.85-year cycle of lunar perigee and covers the 18.6-year nodal astronomical tidal cycle, both of which can influence extreme sea levels (Baranes et al., 2020; Peng et al., 2019; Haigh et al., 2011). The third and final step was to select a semi-random date from this 19-years of tide data that would match with a TC in the baseline and future datasets. The original past/present and future STORM datasets has TCs develop largely between May and November, as occurs in the natural record for this region. Our sub-sets of baseline and future TC data, derived from STORM, all have a simulated month and synthetic year assigned to them. Because of this it was possible to match each TC (surge) to the correct month in the tide data, preserving TC seasonality...*'
* * *
**REVIEWER #2 COMMENTS**

**2 Major comments**

**2.1 Writing and the manuscript structure**

**In my opinion there is a large scope to improve the overall writing of the manuscript. Following is a list of comments -**

**2.1.1.    Introduction was rather hard to follow. For example, the §1 opened with the future surge hazard, then immediately switched to current surge hazard in §2, and then revert back to future surge hazard in §3. My suggestion would be to reorganize it into §2 → §1 → §3.**

**2.1.2.    Two method discussed at L75 and L80 are not same thing it seems. One of them extends data over century, another extends only on decade. The sequence and presentation of these paragraph reads odd.**

**2.1.3.    L91-105: This paragraph is troublesome as it opens with synthetic storm, follows by reanalysis forcing, followed by synthetic data again and STORM dataset etc.**

**2.1.4.    L135-140 should go to a dedicated Dataset section, (Beginning of section 3?) and their properties should be discussed. It is again a bit odd to read about data and method at the end of the Introduction section. Here I was expecting how you are going to structure the manuscript.**

**2.1.5.** **The data/model/discussion section also has similar structure-related issues. I stopped suggesting explicit updates from this point on. I suggest the authors to carefully revise the manuscript with readability (and story) in mind.**

A2.1.(all). Thank you for the valuable feedback for the above 5 points relating to the paper's structure and writing style. We have reorganised the text and structure of the manuscript to improve its readability bearing the above comments in mind. We have added extra headings to distinguish the data and methods sections more clearly (points 2.1.4 and 2.1.5). We have also reorganised sentences and paragraphs throughout the document to make it easier to read, improve clarity and flow more logically (points 2.1.1 – 2.1.3).

**2.2 Modelling and validation**

**2.2.1.** **L146: Is this model already published? If so, please provide past reference.**

A2.2.1. The model was originally created several years ago by our project partners the Southern Institute of Water Resources Research (SIWRR) by our co-investigators Le Quan Quan, Nguyen Nghia Hung and Tran Ba Hoang. The model has been used in various applications by SIWRR but has not previously been published in an English Language International Journal. We also considerably changed the original version of the model by re-configuring the model grid, boundary condition and set up and conducting a more extensive quantitative validation exercise. Hence, this current paper under review here, is the first reference that describes the current model.

**2.2.2.** **How did you choose the model resolution? For a regional study, a resolution of 2.3km (finest) seems rather low, as GSTM 2020 model discussed in the introduction has 2.5km at the coast. Additionally, in most cases the resolution is set to 5-7 km, which is twice as coarser than GSTM.**

A2.2.2. Thank you for your comments – this was an issue raised also by Reviewer #1. The model resolution was determined as a compromise between computational efficiency, model accuracy and simulation run-time. While certainly the GSTM 2020 model is global and would have a greater computational cost at 2.5 km grid resolution than our regional model, we took the position that because we were running almost one hundred thousand simulations of TC movements (when including both baseline and future scenarios), that this range of grid-resolution was the best solution for this task. Additionally, it was a consideration for us that the MIKE 21 FM model should continue

to be used by our partners SIWRR in Vietnam after project completion, on a regular desktop computer.

We have added a few sentences in the discussion section to acknowledge that this resolution does not capture shoreline detail in these locations. The full paragraph reads, *'The model has a variable triangular grid resolution, with greater detail along coastlines. It should be noted that there is potential for sub-optimal accuracy in storm surge levels, particularly within small coastal features such as inlets, bays or estuaries, where coastal resolution is insufficient to capture features in detail. For example, Bertin et al. (2015) showed that within small seas wave radiation can induce set up that transforms storm surge levels along exposed coastlines, with even the small waves entering bays and inlets affecting water levels. Unfortunately, the number of TC simulations entailed in this study meant that there had to be a trade-off between rendering coastal detail and reasonable computation timescales. Future work in such locations, looking at local sections of coastline, would require detailed modelling to estimate extreme sea levels due to storm surge. Additionally, currents and wave action is specifically not incorporated in the modelling as it was outside of the project scope. Such wave models will be a valuable addition to the scientific discussion, as their high spatial resolution at the coastline would ensure that nearshore wave dynamics, such as wave setup, are adequately resolved (Hinkel et al., 2021; Saulter et al., 2017)'*.

The model configuration section has also been updated with the prefix, *'To create a model which could be run around one hundred thousand times, but still offer fair accuracy, we selected grid resolutions at the model's coastal boundaries of ...'*.

**2.2.3. Is any wave model is incorporated? If so which one and how? The resolution issue is further problematic as Wave models needs metric (10s of meter) resolution to properly model the wave setup [1]. In case wave model is not coupled, then wave setup is not computed which can further augment the storm surges [2]. My understanding is that no wave model is coupled. Please provide discussion and adds in the limitations.**

A2.2.3.        No wave modelling was done. We can confirm that our focus was on still water levels to ensure the computational effort and runtimes remained manageable with the scope of the funded project. We have added a sentence to the model configuration section of the manuscript to emphasise this, it says, *'No wave modelling was carried out in this analysis either, since the focus of this paper is to be on still water level'*. Additionally, we have added text to the discussion section to note this limitation. The full paragraph is copied into the response to Comment 2.2.2 above.

**2.2.4.  What do you mean by updated bathymetry in L161?**

A2.2.4.  The bathymetry in the model was updated from the original one provided by our project partners. We utilised a newer SRTM15+ bathymetry which had recently been released by Scripps Institution of Oceanography on their website. Since this is unclear the old sentence '*For bathymetry, an updated 15 arcseconds resolution global dataset from SRTM15+ (v2) was used*' has been replaced with, '*For bathymetry, we replaced the data in the original model with a 15 arcseconds resolution global dataset from SRTM15+ (v2)*'.

**2.2.5.  What does it mean by '2016 tidal model solution' in L167?**

A2.2.5.  The Oregon State University Tidal Inversion Software provides astronomical tidal components for this region. We downloaded the latest output model (dated 2016) for the China seas and Indochina, and used this to create tidal levels for our model boundary locations using their proprietary TMD MATLAB toolbox.

Since it is unclear, the text around this has been modified within the manuscript as follows: '*Tides were only separately modelled for validation and to estimate total sea levels. To simulate tides we generated the astronomical tidal component within the domain using tide data obtained from the Oregon State University Tidal Inversion Software (OTIS, Martin et al., 2009; Egbert & Erofeeva 2002). The harmonic constituents were downloaded from the OTIS web site (http://volkov.oce.orst.edu/tides/) for the seven model open sea boundaries of our model. The data are provided for eight primary (M2, S2, N2, K2, K1, O1, P1, Q1), two long period (Mf, Mm) and three non-linear (M4, MS4, MN4) harmonic constituents. With these data, the tide was then predicted, for each boundary grid point, using the Tidal Model Driver (TMD) MATLAB toolbox (http://polaris.esr. org/ptm_index.html) with the 'China Seas and Indochina region (2016)' tidal model option.*'

**2.2.6.  L181-184: Annual mean sea level values fluctuate, and linearly increase. How do you make it equivalent to model datum (EGM96) by removing a fluctuating value?**

A2.2.6.  Thank you for rising this valid point. Indeed, annual mean sea levels do fluctuate from year to year and have increased over time. However, the data lengths and periods varied greatly among the 27 tide gauge records that we had access to in our region. Some sites only had a year or

two of data. Hence, it was not possible to use a long epoch (for example 18.6 years covering the full nodal cycle) to compute a similar vertical datum at each site, and make this exactly equivalent to the model datum. As a compromise we found that the year covered by the most amount of tide gauge records was 2019. Hence, we simply extracted the annual mean of the tide gauge record from 2019 at each site. Where this year was not available, we took the next nearest year. In the latest version of GTSM (Muis et al., 2020), water levels are referenced to the present-day mean sea level by removing the annual average sea level for each year, and subsequently also by subtracting the mean over the last 19 years from the (de-trended) time series. Ideally, we would have followed the same approach, but as discussed above, the length of several tide gauge records in this region are short. There is a small (few cm) difference between the annual mean of 2019 compared to the mean over the last 19 years. This may cause a small bias in the validation statistics, but at least the bias is consistent among sites. In conclusion we felt this was the best approach, considering the sea level record data length limitations.

**2.2.7. L184-185: Tidal analysis does not remove meteorological influences. At daily scale, for example, S1 tide is forced by diurnal atmospheric pressure loading. At seasonal scale, for example, Sa and SSa components are often influenced by long-term meteorology, freshwater input etc [3].**

A2.2.7.    Thank you for raising this point. We acknowledge that the tidal analysis does not remove all of the meteorological influences, such as the daily scale S1 tide, but it does remove the vast majority of meteorological forcing factors. Importantly, we apply exactly the same tidal analysis procedures to our measured tide from tide gauge records, to the time-series of total water level predicted by the model. Hence, at least these small biases are consistently compared. Furthermore, in the paper we focus mainly on the validation of the main constituents (M2, S2, O1, K1).

**2.2.8. L185-187: If possible please compare for a consistent year.**

A2.2.8.    As we discussed above in regard to point 2.2.6, the 27 records do not all cover the same period, and at some sites, the data only covers a year or two. We choose to focus on January 2019, because this period coincided with when we had the greatest number of complete datasets available across all 27 tide gauge sites. Note, no single year had all 27 records available, so there was always

going to be a compromise. Within the data availability, we do the best we can. Where the data for 2019 was not available, we considered the closest year.

**2.2.9.  Modelled tide and tide gauge data are inherently correlated (L200), hence validation using MAE and correlation fundamentally does not capture the error in the various 'waves' (e.g. tidal constituents). Please use complex error for validation and update Table 2 with complex error instead of MAE. You might also provide a map of the station with Complex Error for the 4 main constituents noted in L214. See for example Tanchant et al 2021 [4].**

A2.2.9.         Thank you for raising this point, we respectfully would prefer to stick with using MAE as this validation method has been used in a wide range of past papers (e.g. Muis et al., 2020 & 2016; Cavaleri et al., 2018; Wahl et al., 2017; Martin et al., 2009). The reviewer mentions that MAE does not capture the error in various tidal constituents; this is why we deliberately validated the total time series, in addition to considering the four main tidal constituents. We therefore don't feel that it is necessary to include complex error as well.

**2.2.10.  L239: is 0.25 degree enough resolution? Why it was chosen?**

A2.2.10.         The 0.25 degree resolution of the imported wind and pressure data from IBTrACS was originally decided on to match the ERA5 data it was to be compared with (Lines 243-245), in the validation exercise. The same template was carried over into the full simulations also (Line 321-322). Increasing the resolution had ramifications that it significantly increased the file sizes of the meteorological forcing datasets, across the 10,000-year storm datasets. We conducted a series of sensitivity tests and found that a 0.25 degree resolution appeared appropriate to capture the meteorological fields. Moreover, within MIKE 21 FM, the meteorological forcing is interpolated (within the software) onto the hydrodynamic flexible mesh, and so it doesn't matter that much what the resolution of the forcing files are, as it is then interpolated to the higher resolution of the hydrodynamic mesh.  Additional text has been inserted into the beginning of this section to make this clearer. This states of the output cyclone wind files generated in MIKE 21 FM; '*This spatial resolution was sufficient to resolve the TC within these forcing files, especially as the wind/pressure files would be further interpolated in the MIKE 21 FM software to the higher resolution of the model mesh as the cyclone traverses through the model domain*'.

**2.2.11. L231-235: A map of these tracks would be useful. This can be added to, for example, figure 2b.**

A2.2.11.   The track of Typhoon Ketsana is shown in the insert in Figure 4b. We totally agree with this comment, and have therefore included the tracks of the other TC events that were used for validation in an inset window within each of the new Supplementary Material figures S1-S3 too.

**2.2.12. L264: Can you show the landfall time in Figure 4? How about other stations? Can those be added as supplementary materials?**

A2.2.12.   Thank you for this comment. We have added a green vertical line to both panels in Figure 4, to indicate when TC Ketsana made landfall (and repeated the same for the other TCs used for validation, in the Supplementary Material figures S1-S3, for other tide gauge locations). Only a handful of TC data could be used for validation as many stations only had short sea level records and captured relatively few landfalling TCs in reality. TC surge levels are generally poorly detected, even between neighbouring tide gauge locations, and so additional station data simply was difficult to obtain for the manuscript.

**2.2.13. I could not find what was the model timestep used for tide, or surge simulation. Please provide this vital modelling detail.**

A2.2.13.   The MIKE 21 FM model set up uses a variable time-step. We have added new text to section 2.2.2 'Hydrodynamic model implementation' to say this. This reads; '*These steering files define the model solution technique, to integrate the time and space variables within the shallow water equations via an explicit scheme, utilising a variable time step interval in the calculation*'.

**2.2.14. I also could not find any detail regarding the model friction formulation. Is it manning? Chazy? roughness length? What is the distribution? Regionally uniform or different? How it was chosen?**

A2.2.14.    Thank you for your comment on this omission. The answer is that a default Manning number was used over the whole domain, with constant value 32 m$^{1/3}$ s$^{-1}$. We continued the text in 2.2.2 'Hydrodynamic model implementation' to now read; '*The recommended default Manning number (i.e., 32 m1/3s-1) was used to define bed resistance over the entire domain*'.

**2.2.15.  Please update figure 2b, with a better colorbar resolving the nearshore bathymetry, which is very critical for storm surge modelling [5].**

A2.2.15.    Thank you for this comment, Figure 2b has been updated with contour lines and a different colorbar – with more detail in the nearshore bathymetry. A comparison between the 'before' and 'after' versions of Figure 2b, is shown in the below Amended Figures and Tables section, for reference.

**2.3 STORM dataset**

**2.3.1.  At several places, Bloemendaal et al. (in review) were referred, and at L292 it was explicitly asked to see a under review paper. The paper was however not accessible. Is it in a closed review journal? If so, I would ask the authors to add substantiating details in the current manuscript. For example, explain in L291 the delta approach used in Bloemendaal et al (under review). Please feel free to use supplementary materials to add further details.**

A2.3.1.    This is an important point and we are therefore pleased to say that the referred-to Bloemendaal et al., paper was accepted for publication on 26th April 2022, in the journal 'Science Advances'. The reference has now been updated in the main text and references list. Here is the link to the new manuscript:

https://www.science.org/doi/full/10.1126/sciadv.abm8438.

**2.3.2.  It is also of interest to know if the seasonal distribution of the STORM dataset follows the observed distribution. Also, if the storm category distribution matches the observed? - or how well they matches.**

A2.3.2.    Bloemendaal et al., (2022) can now provide the detail required, however in response we can confirm that the seasonal distribution of STORM does match the observed distribution. Also, it is shown in Bloemendaal et al., (2022) that the storm category distribution is consistent with observed data in the baseline STORM dataset, whereas within the future dataset there is an increase in the frequency of intense (category 4 and category 5) TCs.

**2.3.3. L304-309: I do not follow why CNRM-CM6 model was chosen? Some linkage with previous study is missing here. Where is this decreasing trend of frequency is found?**

A2.3.3.    There was no particular reason to prefer the CNRM-CM6 Global Climate Model (GCM) over other available GCMs. However, we elected to utilise just one future climate projection in our modelling - it was simply not possible to run all four future GCMs each with 10,000 years of TC activity, within the time-scales of our funded project. In the future we would like to try and run the other three datasets. The fact that the number of more intense TCs (category 4 and category 5) are neither too high or too low in the CNRM GCM, compared to, for example, the extremes of CMCC (Table 3: 9,424 category 5 TCs) or HadGEM (Table 3: 21,661 category 5 TCs) climate models, within the wider Western North Pacific region, was an influencing factor in an otherwise random selection.

Apologies this was not clear, we have updated the text in L304-309 to now read, '*In order to avoid the large computational cost of simulating the storm surge conditions associated with the equivalent of 10,000 years of cyclone activity four times over, we elected to only use data from a single climate model projection. The future STORM dataset based on the CNRM-CM6-1 climate model was selected as - in the range of available climate model options - it ranked in-between extremes in the number of most intense TCs (i.e. category 4 and category 5, Table 3) in the domain, and thus suggested a middle pathway from all the four climate models options*'.

The text relating to decreasing trend of TC frequency was not really helpful in the paragraph alluded to and was creating confusion, therefore it has been removed for clarity. Thank you for this feedback. However to answer, as stated in Bloemendaal et al., (2022), in the future there is projected to be a '*global decrease in annual TC genesis frequency compared to the baseline*' (baseline = present day genesis frequency).

**2.3.4. L310: does that mean there is an increase in frequency along Vietnam coast? and the TC frequency is decreasing in WNP? L343 indicates that for current around 40 year period there is 30,800 cyclones, and for future 35 year period there is 63300 cyclones? The statements in L304 about decreasing frequency, does not make sense with the number presented here. Please explain and clarify. Perhaps it is the selected cyclones, not 'frequency' in the data?**

A2.3.4.        We concur that the statement on TC frequency decreasing in the projected future WNP (Lines 304-306) is confusing in this paragraph and therefore, as also stated in answer to point 2.3.3 above, we have removed it to improve clarity. The point of this section was simply to state that, yes, when comparing similar ~35 year timescales, the projected future TC numbers exceed present day numbers in our model domain and this effects the coastline of Vietnam (and the Philippines) most of all. TC track distributions are also seemingly altered in this region, as illustrated in the Figure 5 hotspot map. This suggests that, by 2050, there is projected to be an uptick in TC activity along all coastlines, but the greatest difference, over this timescale, is around the coastline of central Vietnam (between Hainan Island and southern Vietnam).  This section of text has been updated to hopefully clarify this point.

**2.4 Statistical analysis (Section 3.3)**

**2.4.1. I do not understand what does it mean by "annual maxima approach" was selected? As you explain in Appendix, the STORM dataset has only month information in it, no date, time, or year - is it the case? If so, then how do you decide which storm fall in which year? Needs a clear explanation and logic behind the method chosen here.**

A2.4.1.        The STORM dataset has a synthetic month (weighted by the observed seasonal dataset – see Bloemendaal et al., 2022) and year associated with each created TC. The years are numbered between 1 and 10,000, and while this has no calendar meaning, these years were used to separate a 'year' of TC activity for this statistical analysis. Hence, once we had run all the TC through the hydrodynamic model, we were able to identify the maximum storm surge or water level within each coastal grid cell, for the TC associated with year 1, then year 2 and so on. With excess of 10,000 'years' of data the annual maxima approach is a viable statistical choice, considering that there was no information whether algorithm-generated TCs in any given year-month were independent events (i.e. occurring far enough apart, spatio-temporally, that sequential wind and pressure fields had no residual dependence). We have updated the text in the

manuscript to make this clearer. At the start of section 3.3 the text now reads: '*In estimating these RPLs, we employed the following methodology: (i) the annual surge maxima was found for every one of the 10,000 years of the synthetic record, for each OCP (since each TC in the STORM database has a given synthetic month and year);*'. And in the supplementary section, '*Simulating tides adds the complexity that each cyclone must be randomly assigned a specific day and start time, but STORM only assigns each cyclone to a year and month*'.

**2.4.2. In Figure 8. please provide estimate of confidence interval. For example, bootstrap method can be used for such large dataset. That will also tell us if presenting 10000 year return period value is meaningful.**

A2.4.2. Thank you for your comment. We do not think it is appropriate to include a confidence interval in Figure 8, as it would make the detail in the three lines difficult to make out. Clearly the uncertainty will - and does - increase as you move to higher return periods, which is maximum at the 1000 year return period level. The approach we used to calculate confidence intervals is described in the last paragraph of section 2.2.3 'Computation of return periods'. For each model grid point, we randomly combined the storm surge with a different astronomical tide, repeating the process 100 times. Hence, for each grid point we have 100 different return level estimates, and in the paper we focus on the mean.

**2.5 Discussions**

**2.5.1. A lot of discussion was put forward on bath-tub model, and flooding, but those were neither shown nor discussed in the results section (or appendix, supp.). Detail will be appreciated. For example, sea dykes are discussed in L506, but no information, height, location is presented. Please consider presenting those information - otherwise it is hard to check this claim.**

A2.5.1. Thank you for this useful feedback. We are looking into this in greater detail in a second paper and for this reason we feel that this reference to a bath-tub flood area might not be a useful contribution to the discussion section for this paper after all. Exactly for the reasons that this comment describes. We have therefore removed this text from the manuscript. We are currently undertaking simulations of inundation in the Mekong Delta using DHI's MIKE 11 hydrodynamic model, using our MIKE 21 FM storm surge data at the coastal boundaries, and so will include discussion of this in a future 'following-on' paper.

**2.5.2.  How the storm surge timing differs/similar to river flood timing? Does river discharge has any impact? Does inland flooding can module surge level? Please discuss? Please also indicate that inland flooding is not considered directly in the model.**

A2.5.2.      This is certainly a very interesting point worthy of examination, but unfortunately it is outside of the scope of this study. We are aiming to assess this in the aforementioned second paper, discussed briefly above, coming soon.

**2.5.3.  Please discuss the model resolution issue. (Elaborated above).**

A2.5.3.      This is valuable feedback, thank you. A paragraph of discussion on model resolution and its impact on storm surge levels within small coastal features such as inlets and bays has been added into the discussion section of the paper. This full paragraph has already been copied into the answer for comment 2.2.2 above.

**3. Minor comments**

- **L24: What do you mean by "± tides"?**

With and without tides added. The authors have updated this text around Line 24.

- **L25-26: Repetition of the previous line "storm surge hazard" increases, is essentially stated again in the beginning of the next line. This can be shortened. See comment about "regional amplification" in the Major comments.**

These two sentences have been amended and shortened.

- **L34: This line is a bit jarring to read. I suggest to drop "Because of projected sea-level rise" completely from this line, as you have again said it in L39. Please also re-refer to Kirezci et al. at L39.**

Amended as recommended.

- **L41: The statement that there has been an assumption of stationarity of storm surge statistics needs to be backed up by references. Do you mean the "storm surge" statistics were assumed stationary, or the "storm" statistics were assumed stationary? Needs clarification.**

Line 41 text has been updated and references inserted. Previously there has been an assumption that the properties of storm surges will not change over time, but this has been challenged in recent work. The new text starts with some new references; '*Most past studies have assumed to date that storm surge extreme behaviour has been, and will continue to be, stationary and that the extreme wave climate will change little over large ocean regions (Vitousek et al, 2017; Hinkel et al., 2014). But with projections of a changed climate by the end of this century, this hypothesis has been challenged in recent studies (e.g. Tadesse and Wahl, 2021; Lin-Ye et al., 2020). For European coastlines by 2100, modelling shows that extreme storm surge levels may augment relative sea-level rise by over 30%, under the mean SSP5-8.5 climate projections (Vousdoukas et al., 2016). More recently Calafat et al. (2022) examined 1960-2018 tide gauge observations for north-western European seas and discovered changed trends in surge extremes due to climate variability and anthropogenic forcing'.*

- **L47: "these TC. . . " which one you are indicating? I believe all the TC, not the low-probability ones? is it?**

Yes indeed, all TCs but he more intense TCs are most damaging. The paper has been updated to clarify this; the text now reads, '*It is estimated that currently almost 230 million people around the world are directly exposed to some level of storm surge hazard from either tropical or extra-tropical cyclone activity, based on SwissRe global models (SwissRe, 2017). The populations most acutely at risk from storm surge induced extreme sea levels are those located on low-lying coastlines within tropical zones associated with intense cyclone activity (Dullaart et al., 2021; Edmonds et al., 2020; Kirezci et al., 2020; Woodruff et al., 2013; McGranahan et al., 2007; see also supplementary material section 1)*'

- **L49-51: It is a bit odd to read that 3% of the global population is currently exposed to storm surge hazard, while 2.4-4.1% of global population will be exposed to 1/100 year flood. Are these both quantify the same thing? Does that mean the estimate by one of these two author was over/underestimated? Or do they fall into the error bar of the estimation? This statement also hurt the claim that "it is an increase of 52% compared to today" in L36.**

Thank you for this comment - we can see now how this does seem inconsistent. It is correct that the two sources are quantifying slightly different things and they also estimate their numbers in slightly different ways. Kirezci et al. (2020) is contrasting the present and potential future (SSP5-8.5 climate projections) coastal flooding hazard from tide, surge and wave setup on coastal populations. The calculation of around 128-171 million population currently impacted is drawn from overlaying just the 1:100 year coastal flood extent on NASA's 2015, ~1km grid, Socioeconomic Data and Applications Centre global population dataset. All Kirezci et al., calculations assume absence of any kind of flood defences. A 1km grid indicates some oversimplification of population distributions.

By contrast, SwissRe (2017) has directly measured the impact of *all* storm surges on the global population, regardless of flood levels or return period. The numbers are calculated based on insurance returns within the topical and mid-latitude zones and will naturally take account of flood defence protection or failure. Because these numbers are based on insurance returns there is a possibility that the currently impacted population is overestimated because the same coastal populations could make multiple claims in a given year.

As such, the authors are happy to update these two sections of the paper to reflect the differences of what is being estimated/measured and highlight why the impacted population is slightly different for episodic extreme flood level vs tropical- and extra-tropical cyclone-induced storm surge inundation.

- **L51-56: Would it be possible to shorten this line? Perhaps, it can go to as a table in the appendix, with corresponding death counts, year, affected country(ies), and reference.**

Agreed. This has been removed to the supplementary materials.

- **L60: Please add 'low confidence' interval defined by IPCC.**

Added as described. 'low confidence' is defined as approximately a 2 in 10 (20%) chance of a correct prediction, of how storm surges may contribute to changes in future sea level extremes.

- **L56: Please avoid inconsistency and mismatching. TCs are feature of Tropical regions. For mid-latitude regions they are extra-tropical cyclones or mid-latitude cyclones, not TCs. So, saying "in particular in tropical and mid-latitude regions" is a bit odd.**

Agreed, this text has been updated to reflect the distinction, with thanks for your feedback.

- **L64-65: By capturing do you mean 'observing'/'recording' ? The next lines tells me so, but "capturing" is a bit of odd choice of word. Do you mean to say that infrequent nature of storm surge is the problem, or having a long-term data with necessary spatial resolution is the problem/challenge?**

Yes in these lines, 'capturing' means recording/observing/acquiring/measuring TCs impact on sea levels within short tide gauge record is a problem. The described sentences have been amended and now reads; '*This knowledge shortfall is in large part because of the essential difficulty of assessing tropical-cyclone induced storm surge datasets associated with events that are, by their very nature, somewhat infrequent (Dullaart et al., 2021; Mori et al., 2019). TCs are not only rare events, but they typically affect comparatively short stretches of coastline (<500km) as they approach land, and so storm surges are under-represented in the data from the sparsely distributed network of global tide gauges (Bloemendaal et al., 2020; Pugh and Woodworth 2014).*' In fact, spatial resolution of the TCs themselves within the atmospheric data record is a secondary problem for this science, this issue is discussed later in the paper.

- **L73: "two variant approaches"? → do you mean to say "two variation of an approach" or "two different approach"? Please clarify.**

We have replaced this text with 'two difference approaches'.

- **L79-81: Please add a reference(s) to this claim. (similar to L85)**

References have been added to Lines 79-81 to justify the statement.

- **L83: 'gridded domain' indicates the underlying model is gridded? Is it the case? Although the final distributed output might be 'gridded', GTSM model referred just after is a Delft3D FM implementation - meaning unstructured grid.**

Agreed that this sentence is not 100% clear. Hydrodynamic model meshes can be various shapes to suit the bathymetry and shape of the area to be studied. Only some meshes are in square grid format. This sentence has been amended to now read, '*The second approach involves the use of hydrodynamic models to generate multi-decadal time-series of surge-driven extreme sea levels across oceanic domains.*'

- **L87: At L69 you refers to > 100 year record. But the datasets you are referencing GTSM they are not 100y long? are they? ERA-Interim data starts from 1979, that means GTSM for current climate is only 40 year long.**

Line 69 refers to the ideal length of tide gauge record if anyone wishes to examine extreme storm surge hazard (up to or exceeding 100 years if the most extreme hazard levels are desired). The Line 87, referred to here, relates to Muis et al. (2022) who utilised ERA-interim data with record of ~40 years of wind and pressure data to represent TC activity in the GTSM, and used tide gauge data with 10 years or longer record to calibrate the model. As such the length of record was acknowledged as a limitation of the methodology and their resulting extreme sea levels were confined to a 1:10 year profile.

- **Table 1. Appears at wrong location. Should be moved around when it is first referred.**

Agreed. This can be changed to a better location in the final paper.

- **L148: 'interconnected' is redundant here.**

The authors agree, and have removed this word from this sentence at Line 148.

- **L148-151: Please break these lines.**

Done.

- **L151: What do you mean by 'reconfigured' ? And why it was reconfigured?**

We simply updated the land boundary in the model with a dataset that we could source and reference easily. The previous land boundary was no doubt accurate but we were not able to validate or date it. This word has been changed to clarify this point.

- **L172: Last line is redundant as it is standard.**

We would like to keep this sentence in the document to make it plain that currents are (as is standard), not incorporated. This is for the non-specialist reader.

- **L221: Table 1 should contain data range for each station.**

If this comment refers to a date-range, the authors agree and shall incorporate this data into Table 1

- **L295: drop 'smaller'.**

Done.

- **L296: What do you mean by "uncategorized"?**

Thank you for this comment. This sentence has been updated to explain this distinction: '*To create a sub-set of data for the period 1980-2018 for our own past/present hydrodynamic model we extracted all WNP area TCs that reach at least hurricane strength (Category 1 or greater on the Saffir-Simpson Hurricane Wind Scale; Simpson and Saffir, 1974) from the global STORM dataset (Bloemendaal et al. 2020)*'.

- **L324: What is a 'steering' file?**

Thank you for this feedback. We have updated the text to explain what this refers to. In model terms it is the control file that describes the model set up parameters and points to the correct descriptor/forcing files for a model simulation (e.g. in our case bathymetry, wind, tidal boundary data files). Steering files also control the naming and location of output results files. This text now states, '*MATLAB scripts were also used to create associated steering (control) files. The steering file differentiates the parameters of each simulation, by for example pointing to the next cyclone wind/pressure file or by generating a unique output filename. These steering files define the model*

*solution technique, to integrate the time and space variables within the shallow water equations via an explicit scheme, utilising a variable time step interval in the calculation*'.

- **L393: What do you mean by 'dampens' surge? May be you wanted to characterize opposite way - like, narrow shelf does not allow surge to amplify as much as . . . ?**

The authors feel that 'dampens' is an appropriate word in this instance, but take on board the comment that it lacks clarity and shall review the document to ensure the meaning is conveyed appropriately, and that it refers to the amplitude of the surge wave.

- **Fig6, Section 4.1: It is interesting to see that basically for all the points, 10th largest surge is generated by tracks crossing the point on the west side, except point 1700 - why would be that?**

We wholly agree that this is a very interesting feature of these images. To explain in an oversimplified manner - firstly by characterising the counter-clockwise turning wind fields around the eye of a TC as a simple circular clock face, and split this face into 4 quadrants (12 to 3 o'clock being the first quadrant, 3 to 6 o'clock being the second, 6 to 9 o'clock being the third and 9 to 12 o'clock being the fourth quadrant). And then secondly declaring that 12 o'clock represents the direction of travel of the TC. In Figure 6, with a TC travelling westwards, the rotating wind fields striking onshore for most of the coastline of Vietnam are from the first quadrant of the cyclone, since the coastline faces east. The largest storm surges, located at the red dot in Figure 6, would therefore be found north-east of this TC for most of Vietnam (east for Chinese) coastlines. But around point 1700 the Vietnam coastline is turning and facing westwards. Hence the TC wind fields in the first quadrant are now coming offshore, and it is the third TC quadrant winds now forcing onshore. This flips the positioning of largest storm surges along the Vietnam coastlines from north-east of the TC, to south-west of the TC.

- **L575: Does this line refer to the current study? Or another study?**

Thank you for this comment. We are referring to the state-of-the-art in this scientific field rather than our own study. But as this is not clear we are happy to amend this sentence.
* * *
**REVIEWER#2 References**

[1] Thomas Guérin, Xavier Bertin, Thibault Coulombier, and Anouk de Bakker. Impacts of wave-induced circulation in the surf zone on wave setup. Ocean Modelling, 123:86–97, 2018.

[2] Xavier Bertin, Kai Li, Aron Roland, and Jean-Raymond Bidlot. The contribution of short-waves in storm surges: Two case studies in the bay of biscay. Continental Shelf Research, 96:1–15, mar 2015.

[3] David Pugh and Philip Woodworth. Sea-Level Science: Understanding Tides, Surges, Tsunamis and Mean Sea-Level Changes. Cambridge U. Press, April 2014.

[4] Yann-Treden Tranchant, Laurent Testut, Clémence Chupin, Valérie Ballu, and Pascal Bonnefond. Near-coast tide model validation using GNSS unmanned surface vehicle (USV), a case study in the pertuis charentais (france). Remote Sensing, 13(15):2886, jul 2021.

[5] Y. Krien, L. Testut, A.K.M.S. Islam, X. Bertin, F. Durand, C. Mayet, A.R. Tazkia, M. Becker, S. Calmant, F. Papa, V. Ballu, C.K. Shum, and Z.H. Khan. Towards improved storm surge models in the northern bay of bengal. Continental Shelf Research, 135:58–73, mar 2017.
* * *
**AUTHOR'S ADDITIONAL REFERENCES**

Bloemendaal, N., de Moel, H., Martinez, A. B., Muis, S., Haigh, I. D., van der Wiel, K., Haarsma, R. J., Ward, P., Roberts, M. J., Dullaart, J. C. M., & Aerts, J.. A globally consistent local-scale assessment of future tropical cyclone risk. Science advances, 8(17). 2022.

Calafat, F.M., Wahl, T., Tadesse, M.G. and Sparrow, S.N., 2022. Trends in Europe storm surge extremes match the rate of sea-level rise. Nature, 603(7903), pp.841-845.

Cavaleri, L., Abdalla, S., Benetazzo, A., Bertotti, L., Bidlot, J.R., Breivik, Ø., Carniel, S., Jensen, R.E., Portilla-Yandun, J., Rogers, W.E. and Roland, A., 2018. Wave modelling in coastal and inner seas. Progress in oceanography, 167, pp.164-233.

Hinkel, J., Feyen, L., Hemer, M., Le Cozannet, G., Lincke, D., Marcos, M., Mentaschi, L., Merkens, J.L., de Moel, H., Muis, S. and Nicholls, R.J., 2021. Uncertainty and bias in global to regional scale assessments of current and future coastal flood risk. Earth's Future, 9(7), p.e2020EF001882.

Muis, S., Verlaan, M., Winsemius, H.C., Aerts, J.C. and Ward, P.J., A global reanalysis of storm surges and extreme sea levels, Nature communications, 7(1), pp.1-12, 2016.

Muis, S., Apecechea, M.I., Dullaart, J., de Lima Rego, J., Madsen, K.S., Su, J., Yan, K. and Verlaan, M., A High-resolution global dataset of extreme sea levels, tides, and storm surges, including future projections, Frontiers in Marine Science, 7, p.263, 2020.

Saulter, A., Bunney, C., Li, J.G. and Palmer, T., 2017, September. Process and resolution impacts on UK coastal wave predictions from operational global-regional wave models. In Proceedings of the 15th International Workshop on Wave Hindcasting and Forecasting and 6th Coastal Hazard Symposium, Liverpool, UK (pp. 10-15).

Wahl, T., Haigh, I.D., Nicholls, R.J., Arns, A., Dangendorf, S., Hinkel, J. and Slangen, A.B., Understanding extreme sea levels for broad-scale coastal impact and adaptation analysis, Nature communications, 8(1), pp.1-12, 2017.
* * *
**Amended Figures and Tables:**

[Figure]

*Figure S1 - Validating modelled surges using ERA5 (red dashed) and Holland Model using IBTrACS (red dotted) wind and pressure fields against measured data (blue): Typhoon Sally surge at tide gauge 5: Zhapo, China (inset or see Figure 1 for location), located closest to Zhapo station in the early hours of 9th September 1996 (green vertical line). Firstly (a) comparing total sea levels, and then (b) comparing surge-only water levels.*

*Table S1 - The Root Mean Square Error (m) between (a) measured tide gauge) and modelled total water levels and (b) tide-removed measured data and modelled surge-only water levels from all the validation hindcast simulations.*

| Typhoon name (date) | (a) Total Water Level RMSE (cm) | | (b) Surge-only Water Level RMSE (cm) | |
|---|---|---|---|---|
| | ERA5 data | IBTrACS data + Holland model | ERA5 data | IBTrACS data + Holland model |
| Sally (September 1996) | 33 | 30 | 20 | 15 |
| Linda (November 1997) | 41 | 28 | 39 | 20 |
| Mangkhut (September 2018) | 42 | 20 | 36 | 15 |
| Ketsana (September 2009) | 40 | 18 | 34 | 10 |

[Figure]

*Figure 1 - Track density = the number of Saffir Simpson category 1-5 Tropical Cyclone tracks passing through each 0.5 degree x 0.5 degree grid cell within each ~35 year period. Left: Baseline STORM track density of Saffir Simpson Category 1+ ( i.e. excluding Tropical Storms), Middle: CNRM climate model- Future STORM track density, of Saffir Simpson Category 1+, and Right: The cyclone track density difference between them.*

[Figure]

*Comparison of Figure 2b- Left is original, Right is updated bathymetry with contours and new colorbar. [this is SRTM15+ ocean bathymetry for our South China Sea domain. The 250m isoline is highlighted in red]*

---

## Author Response (AR2)

Dear Francisco Campuzano, Guest Editor for Natural Hazards and Earth System Sciences,

Thank you for passing on the thoughtful comments from the two reviewers. We have carefully considered each comment in turn, and below, provide a detailed response to each point raised. We believe this paper has been greatly strengthened and we look forward to hearing back from you.

Regards,

The authors
* * *
**Reviewer 3: Major revisions**

1. The paper needs major actions to properly describe overall objectives and adopted methodology.

Thank you for your comments. We have re-written large parts of the introduction section to make the motivation of the study stand out more clearly and to better articulate the aims and objectives of the manuscript. For example, we have re-written the first paragraph in the introduction regarding the impact of tropical cyclone (TC) events. This paragraph now reads:

> *Around the world's coastlines it is estimated that ~230 million people are directly exposed to some level of storm surge hazard from either tropical or extra-tropical cyclone activity (SwissRe, 2017). The populations most acutely at risk from storm surge induced extreme sea levels are those settled on low-lying coastlines within tropical zones associated with intense tropical cyclone (TC) activity (McGranahan et al., 2007; Woodruff et al., 2013; Kirezci et al., 2020; Edmonds et al., 2020; Dullaart et al., 2021). Yet global assessments of flood risk regularly overlook the contribution of low probability TC events when considering storm surge induced extreme sea level flooding (Muis et al., 2016; Dullaart et al. 2021). Furthermore, while there is considerable uncertainty regarding future changes in TC intensity and frequency, particularly at a local scale, it is thought that the risk of TC induced storm surge flooding will increase in the future (Bloemendaal et al., 2022). To better protect present and future coastal communities, it is vital that we improve our understanding of local-scale TC driven storm surge hazard and risk.*

We have also largely re-written the last two paragraphs of the introduction section to describe the overall aim of the paper and specific study objectives more clearly. We have listed an overall aim and three specific objectives. The final two paragraphs of the introduction now read:

> *The specific objectives and structure of the paper is as follows. As a first objective, we configure a depth averaged hydrodynamic model of the South China Sea and extensively validate it against measured sea level data from tide gauges in the region. A description of the hydrodynamic model and validation exercise is provided in Section 2. As a second objective, we force the hydrodynamic model with 10,000 years of TC activity for the past/present (1980-2017) and future (2015-2050; based on a high-emission scenario), from the novel STORM synthetic dataset of Bloemendaal et al. (2020, 2022). From the model outputs, we estimate both past/present and future storm*

*surge and extreme sea level probabilities along the coastlines of south China, Vietnam, Cambodia, Thailand, and Malaysia. The approach we take to simulate the storm surges and calculate the associated extreme probabilities is described in Section 3. As a third objective, we compare the past and future probabilities, first for just the storm surge component and then for total water levels (i.e., storm surge plus astronomical tide). The results of this comparison are described in Section 4. As a sub-objective, we also examine the tracks of the cyclones that are responsible for generating the largest storm surges in particulate locations along the coastline of the case study area. The key findings are discussed in Section 5, and conclusions are given in Section 6. Note, wave set up and run up are an important contribution to extreme sea levels, particularly in regions of intense TC activity; however, mainly for simplicity and due to the computational expense of running many tens of thousands of model simulations, we focus here only on storm surges and still water levels.*

For your comment about the adopted methodology, we address this below in regard to point 4.

2. The abstract is quite general and should highlight the main topic and messages that the paper is expected to address.

Thank you for your comment. We agree the abstract read quite general before and didn't highlight strongly enough the motivation of the study and key findings. We are confined, however, by the recommended abstract word-limit for this journal, which is 100-200 words. However, we have reworked the abstract to highlight the main topic and findings and it now reads as follows:

*Coastal floods, driven by extreme sea level, are one of the most dangerous natural hazards. The people at highest risk are those living in low-lying coastal areas, exposed to tropical cyclone forced storm surges. Here we apply a novel modelling framework to estimate past/present and future storm surge and extreme sea level probabilities along the coastlines of south China, Vietnam, Cambodia, Thailand, and Malaysia. A regional hydrodynamic model is configured to simulate 10,000 years of synthetic tropical cyclone activity, representative of a past/present (1980-2017) and a high-emission scenario future (2015-2050) period. Results show that extreme storm surges, and therefore total water levels, increase substantially in the coming decades, driven by an increase in the frequency of intense tropical cyclones. Storm surges along the south China and northern/southern Vietnam coastlines increase by up to 1 m, significantly larger than expected changes in mean sea level rise over the same period. The length of coastline that is presently exposed to storm surge levels of 2.5 m or greater more than doubles by 2050. Sections of Cambodia, Thailand and Malaysia coastlines are projected to experience storm surges (at higher return periods) in the future, not previously seen, due to a southward shift in tropical cyclone tracks. Given these findings, coastal flood management and adaptation in these areas should be reviewed for their resilience against future extreme sea levels.*

3. "Introduction" is quite ok, but please try to complete the references (for example, no references for ERA-Interim and ERA5).

Thank you for your comments. We have carefully looked through the manuscript. We only refer to ERA-Interim once on line 138-140, when describing the Global Tide Surge Model (GTSM) which was forced with ERA-Interim. We previously did not provide a reference for ERA-Interim, because the focus of the sentence was on the Muis et al. (2016) paper that applied the GTSM model rather than on the ERA-Interim dataset itself. However, we have now

reworded the sentence, so it also includes the reference to the Dee et al. (2011) paper that describes the ERA-Interim reanalysis. The sentence now reads:

> *An earlier version of the Global Tide and Surge Model (GTSM; using ERA-Interim reanalysis data – see Dee et al., 2011) was found to underestimate TC induced extreme sea levels for this reason.*

We have also revised sentence 139-141 which include reference to ERA5. We note again here that the focus on the paper is on the Muis et al. (2020) study and not ERA5. However, again, we have reworded the sentence to include the C3S (2020) paper that describes ERA5. The sentence now reads:

> *This problem was subsequently improved in the latest GTSM iteration, v3 (Muis et al. 2020), with an updated model resolution and use of the ERA5 reanalysis meteorological dataset (Hersbach, 2020). The authors successfully simulated past/present storm surges and extreme sea levels.*

4. "Data and methods": I would suggest to reorganize it. The hydrodynamical model requires a more schematic description: the area of study description is missing. Many references to some MATLAB codes but without going into details on the specific implemented method makes this section quite weak.

Thank you for your helpful comment here. We regret the methods section was not as clear as it could have been. We have therefore re-written and reorganised section 3, now entitled 'Approach for simulating present and future extreme sea levels'. Also, upon the recommendation of the other reviewer (see our response to Review 4, point 2, below) we have now included a flow chart within the supplementary material (also shown below) that outlines the configuration steps and datasets. We believe this has significantly improved this section.

Regarding the model study area specifically, we have written a new paragraph, as follows:

> *The model grid we created for the South China Sea is shown in Figure 2a. It extends from approximately 100 degrees W to 120 degrees W and 3 degrees S to 25 degrees N. It encompasses the east coast of Malaysia and Thailand, the entire coast of Cambodia and Vietnam, and extends to South China. We developed the model off the continental shelf of these countries with the eastern domain of the model running along the coastline of Borneo Malaysia, Brunei, the Philippines, and Taiwan (Figure 1). The model grid has seven open sea tidal boundaries, shown in red on Figure 2a. The model mesh reduces from ~52 km at the open sea boundaries to approximately ~2 km along parts of the coast. Along our study coastline (the grid cells shown in green in Figure 2a), the model mesh reduces in resolution from ~11 km along south China and Malaysia to ~5 km around Hainan Island and along the coast of Thailand and China, down to ~2 km along the Vietnam coastline (as mentioned above, we focus in on Vietnam, as this is the central region of interest in the project that funded this study). We are aware that some global hydrodynamic models (such as GTSMv3; Muis et al. 2020) now have a resolution along the coast of finer than 2 km, and we could have increased the coastal resolution further. However, we purposely didn't go to a finer resolution because: (1) our study involved running the model almost one hundred thousand times for synthetic cyclones - increasing the resolution would significantly increase total run-time; and (2) our model design should consider our Vietnamese co-authors' objective to potentially use the model for future studies, without having regular access to a super-computer. Therefore, we wanted the model to be easy to run on a standard desktop computer.*

[Figure]

*Figure S1 – The basic model configuration*

[Figure]

*Figure S2 – Model validation. The storm surge validation process is described in section 2.3 of the paper; we use different wind and pressure input data to simulate historic TC events, using (1,a) ERA5; (1,b) Holland formula or (1,c) no meteorological forcing data. Separately, validation of astronomical tides is illustrated in schematic (2,d) with a tides-only model simulation, as described in Section 3 below, and in section 2.2 of the paper.*

[Figure]

*Figure S3 – MIKE 21 hydrodynamic model configuration for (3) past/present scenario; the (4) future scenario and (5) testing the model sensitivity to tide-surge interactions (see Section 3).*

Regarding your comment about MATLAB, we now only refer to MATLAB code on two main occasions in this paper. First when referring to the TMD toolbox for creating the open tidal boundary conditions, and second for the T-TIDE harmonic analysis. For both of these we provide web-links, so it is completely transparent where the code comes from, and we have added more complete descriptions of what the code does.

5.  References to "project partners" are given in the section, please clarify the framework where the contribution has been designed.

Thank you for your comment. We agree the reference to project partners was confusing. For your context, the project that funded this study involved researchers from the University of Southampton, UK and researchers from the Southern Institute of Water Resource Research (SIWRR) and funding to undertake the study was received from both the UK and Viet Nam research councils. The project partners we were referring to were SIWRR. We have removed all reference to project partners in the paper to avoid any confusion. The author list and acknowledgments make it clear this project was a partnership between UK and Vietnamese researchers, so it is not necessary to refer to project partners in the paper.

6.  It is important to provide quantitative description on the approaches that You decided to propose: for example, in ln. 254 how do You generate the wind and pressure modified fields? There are many cases like this in the section, so please try to focus on the effective method without being too general.

Again, thank you for your thoughtful comment. We have now completely revised the previously named Section 2: 'Model configuration and validation'. When validating how well the model predicts storm surge and total water levels we use wind and pressure fields derived in two ways. First, we used *u* and *v* direction component wind, and mean sea level atmospheric pressure, fields from the ERA5 reanalysis data set. However, it is known that ERA5 does not fully capture the intensity and track of TCs, due to its spatial resolution. Hence secondly, we also derived a set of meteorological fields for each of the four chosen storms using the empirical approach of Holland (1980). In this case we used the Cyclone Wind Generation toolbox (DHI, 2017b) built into MIKE 21. We have added the following paragraph to Section 2.3 to clearly describe what we have done, and included Figures S2 and S3 flow charts in the Supplementary Materials (shown in answer to comment 4 above):

*To simulate these four TC events, the second step was to create spatially and temporally varying wind and atmospheric pressure fields to force the hydrodynamic model. We forced the model with two different meteorological fields and compared the results. First, we used u and v wind and MSL atmospheric pressure fields directly from the ERA5 reanalysis data set (Hersbach et al., 2020). These were downloaded from the Copernicus climate data store (https://cds.climate.copernicus.eu/), for the known cyclone dates, on a regular 0.25° x 0.25° grid at hourly resolution (Hersbach et al., 2020). These data were simply clipped to the area of interest and imported into a MIKE 21 FM grid file format with no other modification. As discussed in the introduction, ERA5 may not accurately capture the intensity and track of TCs, due to its spatial resolution. Hence, we also derived a second set of meteorological fields for each of the four chosen storms. In this instance, we derived spatially and temporally varying wind and atmospheric pressure fields using the empirical approach of Holland (1980). To do this, we used the Cyclone Wind Generation toolbox (DHI, 2017b) built within MIKE 21 FM. To generate the empirical wind and pressure fields we inputted the track, of each of the four selected TCs at 3-hourly timesteps, as captured in the IBTrACS database, along with central atmospheric pressure and radius to maximum wind values. We selected the 'Single Vortex Holland' option within the toolbox, which creates an estimate of the Holland B parameters using the Holland Formula specified in Harper and Holland (1999). This tool therefore generated u and v wind, and pressure files, for each of the four TC events, on a 0.25° x 0.25° grid resolution to match ERA5 spatial grid resolution, for fair comparison.*

We have also added the following paragraph to Section 3.2, describing the model implementation:

*We now describe how we forced the model with wind and pressure fields derived from these 30,843 baseline and 63,328 future TCs. We generated spatially and temporally u and v wind and atmospheric pressure fields, using the approach described above in Section 2.3. We used the MIKE 21 FM Cyclone Wind Generation toolbox, inputted with the track of each synthetic TC at 3-hourly timesteps, along with central atmospheric pressures and radius to maximum winds values, obtained from the STORM dataset. Again, we selected the 'Single Vortex Holland' option, with Holland B parameters estimated using the Holland Formula specified in Harper and Holland (1999), and generated u and v wind and pressure files on a 0.25° x 0.25° resolution grid. This spatial resolution was sufficient to resolve the TC within these forcing files, especially as the wind/pressure files would be further interpolated in the MIKE 21 FM software to the higher resolution of the model mesh as the cyclone traverses through the model domain.*

7. "Results" deserves a major attention: the model results and validation exercises need to be reorganized and the text should be rephrased in a way the reader can understand what the figures are showing.

Thank you for this comment. Firstly, we have now re-written the section describing the tidal and storm surge model validation (Sections 2.2 and 2.3). We believe this provides a much clearer overview of the suitability of the model to accurately predict tides, storm surges and total water level across the study region. Secondly, we have also re-written the results section.

Again, we believe this rewriting and restructuring of the results, greatly improves the readability of this paper.

8. Consequently, "Conclusions" might need for revisions in order to summarize and provide the key messages the contribution is expected to provide.

Once again thank you for your comment. We have revised the conclusions section, to bring out the key messages much more clearly. The conclusion section now reads:

> *As the latest IPPC report has indicated (Fox-Kemper et al., 2021), there is currently little (~20%) confidence in the scientific community being able to accurately predict future changes to storm surge characteristics, particularly in regions of the world exposed to TCs. The low level of confidence arises both because of the significant challenge of predicting changes in TC activity at a local and regional scale, and because relatively few studies have assessed changes in storm surge driven by TCs. Therefore, our overall aim in this paper is to apply a novel modelling framework to more accurately estimate both present and future storm surge and extreme sea level hazards, by considering the densely populated coastlines of south China, Vietnam, Cambodia, Thailand, and Malaysia as a case study.*

> *We configured a depth averaged hydrodynamic model of the South China Sea and extensively validate it against measured sea level data from tide gauges in the region. We then forced the hydrodynamic model with 10,000 years of TC activity, representative of a past (1980-2017) and future (2015-2050) future period, based on a high-emission climate projection scenario. From the model outputs, we estimate both past and future storm surge and extreme sea level probabilities along the coastlines of south China, Vietnam, Cambodia, Thailand, and Malaysia.*

> *Our results showed that extreme storm surges, and therefore total water levels, increase substantially in the coming decades (up to 2050), under a high emission (SSP5-8.5) scenario, driven by an increase in the frequency of intense TCs. The increases in storm surges in some regions, e.g., along the south China, and northern and southern Vietnam coastlines, can exceed ~1 m in the 1% AEP measure; significantly more than the expected changes in mean sea level rise over this period. The length of coastline that is currently exposed to storm surge levels of 2.5 m or greater more than doubles (353 km to 930 km), between the baseline and future high emission scenario. Around the low-lying and densely populated areas of the Red River and Mekong Delta storm surges with an AEP of 1% (1 in 100 year return period) today, are likely to see a change in frequency to ~3% AEP (1 in 30 year return period) over the coming decades. Furthermore, at higher return periods, the coastlines of Cambodia, and parts of Thailand and Malaysia are predicted to experience storm surges induced by TCs in the future scenario, whereas presently they don't. A similar methodology to that applied here could be used to assess changes in storm surges and extreme water levels in other regions of the world that are exposed to TC activity.*

> *Many future projections of extreme sea level, at global, regional, or local scales, only account for changes in relative mean sea level, but here we have shown that changes in storm surges could be significant, and even exceed changes in MSL in some areas. Our study area has many low-lying and densely populated coastlines, such as the major river deltas in this region, that are especially vulnerable to storm surges. Given these findings, coastal flood management, planning and adaptation in these areas should be*

*reviewed for their resilience against changes in storm surges and total water level levels in the future.*
* * *
**Reviewer 4: Minor revisions**

1. The model established in this paper considers stationary sea level and does not take sea waves into account, which is somewhat different from the actual situation. Therefore, the impact of this practice on the research results should be explained in more detail.

Thank you for raising this good point. In this study we focus only on storm surges and still sea level and ignore the effect of waves. We agree that waves, particularly wave setup and runup, are very important contributors to extreme sea levels. However, we did not include the effect of waves because, within the limited funding and scope of the project it would have been too costly to also configure a wave model and it would have been too computationally expensive to run the many thousands of additional simulations with waves. We have added a sentence at the end of the introduction section to acknowledge the importance of waves, but stress that this paper focus only on storm surges and still sea levels. The new sentences reads:

> *Note, wave set up and run up are an important contribution to extreme sea levels, particularly in regions of intense TC activity; however, mainly for simplicity and due to the computational expense of running many tens of thousands of model simulations, we focus here only on storm surges and still water levels.*

We have also added the following paragraph in the discussion:

> *We also highlight again that waves, particularly wave set up and run up, are important contributors to extreme sea level and coastal flooding, particularly in areas of intense TC activity. Due to project time, computation outlay, and limit of budget that funded this study, we have focused in this paper only on storm surges and still sea levels. Future work could include waves. The same framework could be applied to simulate past/present and future wave climates and incorporate them in estimates of total water level probabilities.*

2. The important contribution of this paper is the establishment of the hydrodynamic model. It is suggested to add the flow chart of the establishment of the model, which is helpful to present a more intuitive and comprehensive model establishment process and results.

Thank you for your comments. We have comprehensively re-written Section 2, which describes the model configuration and validation. We have added lots more detail in places. Regarding the flow chart, this is indeed a good comment. We have added a flow chart schematic figure in the Supplementary Material. This shows the processes of: (S1) the basic model configuration; (S2) tidal/surge validation; and (S3) the 10,000 year simulations for past/present and future scenarios.

[Figure]

*Figure S1 – The basic model configuration*

[Figure]

*Figure S2 – Model validation. The storm surge validation process is described in section 2.3 of the paper; we use different wind and pressure input data to simulate historic TC events, using (1,a) ERA5; (1,b) Holland formula or (1,c) no meteorological forcing data. Separately, validation of astronomical tides is illustrated in schematic (2,d) with a tides-only model simulation, as described in Section 3 below, and in section 2.2 of the paper.*

[Figure]

*Figure S3 – MIKE 21 hydrodynamic model configuration for (3) past/present scenario; the (4) future scenario and (5) testing the model sensitivity to tide-surge interactions (see Section 3).*

3. The conclusions of this study can provide a basis for policy making in coastal areas. It is suggested that this study should add specific management and prevention measures to the high risk areas of storm surge predicted by the model, which will further improve the value of this study.

Thank you for your comment. However, we believe it is beyond the scope of the study to go into detail about the specific management and prevention measures. We are currently working on a follow-up paper for the Mekong Delta, which will utilise the outputs of this manuscript, and aim to discuss more detail of coastal management measures there. However, we have added a paragraph to the end of this manuscript to comment on coastal management, planning and adaption. It now reads:

> *Many future projections of extreme sea level, at global, regional, or local scales, only account for changes in relative mean sea level, but here we have shown that changes in storm surges could be significant, and even exceed changes in MSL in some areas. Our study area has many low-lying and densely populated coastlines, such as the major river deltas in this region, that are especially vulnerable to storm surges. Given these findings, coastal flood management, planning and adaptation in these areas should be reviewed for their resilience against changes in storm surges and total water level levels in the future.*

4. The future STORM scenario which uses a mean SSP5-8.5 profile adopted in this paper is the scenario with the highest emissions, which is the emission scenario without policy intervention. Now, however, there is an international agreement to control carbon emissions, and various countries, especially China, are taking steps towards carbon peaking and carbon neutrality. Therefore, in this context, whether ssp585 has predictive significance, in addition, according to the model in this paper, whether the emission of low intensity, such as ssp119, ssp126, etc., will get significantly different or even opposite conclusions? I am worried about this.

Thank you for this valid point. First, we wish to point out that the STORM model dataset for the future, containing the 10,000 year of synthetic cyclones and developed by Bloemendaal et al. (2022), was only done for the SSP5-8.5 future climate scenarios. Bloemendaal et al. (2022) did not produce synthetic cyclones for any other climate scenario, so we were only able to run

the datasets that were available. Second, we do agree that this represents a low-probability/worst-case scenario for a 2100 climate. Nevertheless SSP5-8.5 might also perhaps represent a caution of where we could all be if efforts to control carbon emissions fail in the next several years and decades. It is also worthwhile pointing out that our results, based on a SSP5-8.5 scenario for 2050, could help inform future flood defence policy in this region. As such, with so much uncertainty at the current time in our future climate pathway, then flood defence policy to a 'worst-case' climate scenario could be a risk-averse option to aim for when protecting the region's coastlines. We have inserted an extra paragraph in the discussion section to explore why a SSP5-8.5 future climate might be a good choice for mid-century estimates of weather patterns for our flood model. The new paragraph reads:

> *Finally, we stress that in this paper we have utilised the future STORM database from Bloemendaal et al. (2022) that is based on a SSP5-8.5 climate change scenario. Note, Bloemendaal et al. (2022) only created TC for the SSP5-8.5 scenario and no others, so we were not able to run other climate projections. SSP5-8.5 assumes a future society that has developed within the highest greenhouse gas emissions pathway, being more energy-consumptive but also successfully using innovation and technology to adapt, rather than mitigate, against its environmental problems. It has less social and economic inequality compared to most other pathways and has a booming global economy. A SSP5-8.5 future represents a low-likelihood outcome for 2100 (Hausfather and Peters 2020; IPPC 2022; Pielke et al., 2022) and that the most plausible scenario for 2100 is thought to be closer to SSP2-4.5 and SSP3-7 if pledges and global climate progress so far are incorporated. Nevertheless, the carbon gap between what we have now and what we ought to achieve by 2050 looms large (Hausfather and Peters 2020; Pielke et al., 2022). It will require an enormous global effort to achieve the policy goal of global net-zero CO2 emissions by 2050, not least because this policy relies on decarbonisation technologies for decades to come (IEA, 2022; Pielke et al., 2022). Climate projection outcomes to 2100 are provisional, which means our climate by 2050 is unknown. This uncertainty is also seen in the International Energy Agency's (IEA) research which suggests that populations in 2050 would be living through an intermediate era, with CO2 emission levels that have plateaued before levels dip further by 2100 (Lee et al., 2021; IEA, 2022; Pielke et al., 2022). Indeed, Schwalm et al. (2020) argues that the SSP5-8.5 scenario is, in fact, the best tool to quantify physical climate risk by 2050, because this scenario has so far most closely tracked the total cumulative CO2 emissions to date. Nevertheless, whether SSP5-8.5 represents the future worst case in global emissions by mid-century, or whether it truly characterises the mid-point of global climate on the path to best-case outcomes, SSP5-8.5 outputs have real value when attempting to define future storm surge flood risk response. We therefore believe the model outcomes in this paper provide useful data not only for decision-makers tasked with developing flood policy to serve future generations, but also for sectors thinking how to develop flood defences that age well because they are capable of withstanding the worst-case-scenario storm surges of the future.*

---

## Author Response (AR3)

Dear Editor,

Thank you for passing on the further useful comments from two reviewers. We have considered and implemented each comment. Below we respond to each point raised. We believe this paper has been greatly strengthened and we look forward to hearing back from you soon.

Regards,

The authors

**- Reviewer 2 -**

**Major comments:**

L400/L645/L735: This finding of shifting storms is very interesting. I understand that it was chosen to run (hydrodynamic model) only for CNRM-CM6-1 derived storms for future climate. But I will suggest to do similar analysis as shown in Figure 5 for all other available storm tracks (not necessary to add in the main figure, but consider as supplement if possible) to be more confident about this "shifting storm" trend.

We can confirm that the shifting of TC (and TC-forced storm surge) activity within the South China Sea region is also observed within other climate model track data. Below we show: (a) the original CNRM-CM6-1 climate model results, alongside; (b) EC-Earth3P-HR; (c) HadGEM3-GC31-HM; and (d) CMCC-CM2-VHR4 climate model TC track densities. All show the same impact, to differing degrees, at the Vietnam coastline. We inserted (b-d) of the figure below as a new figure within Appendix 1 (section 4, Figure S8).

**(a) CNRM-CM6-1 Climate Model - TC track density**

[Figure]

**(b) EC-Earth3P-HR Climate Model - TC track density**

[Figure]

**(c) HadGEM3-GC31-HM Climate Model - TC track density**

[Figure]

**(d) CMCC-CM2-VHR4 Climate Model - TC track density**

[Figure]

2. EVA/Figure 8: I understand that having large number of storm sample immediately reduce the error-bar in the EVA analysis, but I would prefer to have that estimate in the figures (as shaded regions). For example, see study by Leijnse et al. (2022). This is particularly important considering that the error-bar/range associated with sea level rise projection is already quite large.

We have added confidence bounds to Fig 8 showing the total water level return periods. We have 10,000 years of annual (block) maximum storm surge levels, at every point location along the coastline. To better understand the uncertainty inherent in the tide data, for each of these annual maximum storm surge data, we generated a new sample set of total water levels by adding 100 random tides (from a 19-year period -encompassing a complete lunar perigee and lunar nodal cycle) to storm surge level, as described in the manuscript. The solid return period lines in Fig.8 were calculated based on the mean of this 100-sample set. New shaded areas are now inserted into Fig.8 to also display the 95% confidence intervals associated with this mean value. Red shading indicates the confidence bounds for the past/present (baseline) total water levels at each return period, and green shading indicates the confidence bounds for the projected future total water levels by year 2050. In addition, the relative mean sea level rise (SLR) of 0.25 m for this coastline by 2050 has confidence bounds at 0.17 m and 0.35 m (the 17 and 83 percentile range, respectively), in the latest IPCC 6[th] Assessment report (SSP5-8.5 scenario)**, and this additional uncertainty was added to the future total water level return period + SLR dashed line in each figure (blue shading).

[Figure]

Figure 8 - The relationship between past/present baseline (red) and future (green) total water level return period (Log scale, 1:X years) at equidistant locations at and around the Vietnam and South China coastline. Future return periods with 0.25 mean sea level rise, due to climate change by 2050, is shown with a dashed blue line. Shaded areas indicate the 95th percent confidence level around each mean total water level return period value.

\*\* Median projections of global and regional sea level rise, relative to a 1995-2014 baseline. Data was accessed on 19.04.2023 using the Sea Level Project Tool: https://sealevel.nasa.gov/ipcc-ar6-sea-level-projection-tool

**Minor comments:**

- Wave contribution: Please refer to Melet et al. (2018) and Dodet et al. (2019). L92 is ok, but L152-L155 Could be removed or can go to the corresponding discussions.

We have incorporated these two new references on wave set up and wind waves, with thanks, and removed lines 152-155, which outline exclusion of wave set up and run up in the model, to avoid repetition.

- L212: Please consider providing the values for default configurations in MIKE21 FM, either in Appendix or in supplement. That will make the study a bit more reproducible (particularly with other models).

We have added an extra table to Appendix 1 (Table S1) with more details of the model set up values, including default values.

- I could not find any mention of model timestep in the manuscript? Is it mentioned somewhere that I missed or it is missing? Please incorporate this valuable modelling info.

Thank you for raising this point, as it is an important consideration in model recreation. To answer, we set an initial overall discrete timestep interval of 3600 seconds in our MIKE 21 FM model. All time steps within the simulation for the various modules are synchronized to this overall discrete time step. Our simulation was run within the hydrodynamic module of MIKE 21 FM. This module used a variable timestep to resolve the shallow water equations with the benefit that the timestep will dynamically adapt to stabilize the model simulation as it progresses. We used a minimum timestep of 0.01 seconds and maximum timestep of 25 seconds (with critical CFL number of 0.8) as bounding limits. This allowed us to create the most stable model possible - given the huge range of extreme wind and pressure fields it had to handle over such a large domain. Furthermore, the time integration and space discretization timestep limits were identically chosen – both following a higher order integration method - to produce a more accurate end result also. We have updated the flow diagram in Appendix 1 to include the model timestep information, and added additional text into Section 2.1 (Model configuration) as below:

> "*The overall discrete model timestep was set to 3,600 seconds (i.e., 1 hourly), however the MIKE 21 FM hydrodynamic module which resolves the shallow water equations over the model domain uses a variable timestep to ensure stability of the model during the simulation. This means that timesteps would reduce further (between 0.01 and 25 seconds)*

*as necessary, between outputs, to ensure optimal time integration and space discretization solutions.*"

- Using Mean Absolute Error, Correlation Coefficient is not a good idea to validate the tide (more precisely the tidal constituents) as they are coupled amplitude and phase (Although using MAE and CC is very common in the literatures). For example two M2 waves separated by minor phase shift will have high MAE (e.g. high error), but also will show high correlation. More useful and compact metric for validating tide is complex error - see Tranchant et al. (2021) for an example. As for the current manuscript, two different tables, and figures are provided with various information - I am accepting it as is. However, please consider using complex error or similar complex metric for tidal validation in the future to be more compact and precise.

Thank you for this advice, the issue of also incorporating complex error analysis is certainly something we will aim to do in future manuscripts, for the good reasons you kindly provide.

**References**

- Dodet G, Melet A, Ardhuin F, Bertin X, Idier D, Almar R (2019) The Contribution of Wind-Generated Waves to Coastal Sea-Level Changes. Surveys in Geophysics 40:1563–1601

- Melet A, Meyssignac B, Almar R, Cozannet GL (2018) Under-estimated wave contribution to coastal sea-level rise. Nature Climate Change 8:234–239

- Tranchant Y-T, Testut L, Chupin C, Ballu V, Bonnefond P (2021) Near-Coast Tide Model Validation Using GNSS Unmanned Surface Vehicle (USV), a Case Study in the Pertuis Charentais (France). Remote Sensing 13:288